# Hyperparameter Optimization via Interacting with Probabilistic Circuits

**Jonas Seng**[1,*]  **Fabrizio Ventola**[1,*]  **Zhongjie Yu**[1]  **Kristian Kersting**[1,2,3]

[*]Equal contribution.
[1]Computer Science Department, Technical University Darmstadt
[2]Hessian Center for AI (hessian.AI)
[3]German Research Center for AI (DFKI)

**Abstract**  Despite the growing interest in designing truly interactive hyperparameter optimization (HPO) methods, to date, only a few allow to include human feedback. Existing interactive Bayesian optimization (BO) methods incorporate human beliefs by weighting the acquisition function with a user-defined prior distribution. However, in light of the non-trivial inner optimization of the acquisition function prevalent in BO, such weighting schemes do not always accurately reflect given user beliefs. We introduce a novel BO approach leveraging tractable probabilistic models named probabilistic circuits (PCs) as a surrogate model. PCs encode a tractable joint distribution over the hybrid hyperparameter space and evaluation scores. They enable exact conditional inference and sampling. Based on conditional sampling, we construct a novel selection policy that enables an acquisition function-free generation of candidate points (thereby eliminating the need for an additional inner-loop optimization) and ensures that user beliefs are reflected accurately in the selection policy. We provide a theoretical analysis and an extensive empirical evaluation, demonstrating that our method achieves state-of-the-art performance in standard HPO and outperforms interactive BO baselines in interactive HPO.

## 1 Introduction

Hyperparameters crucially influence the performance of machine learning (ML) algorithms and must be set carefully to fully unleash the algorithm's potential [Bergstra and Bengio, 2012, Hutter et al., 2013, Probst et al., 2019]. *Hyperparameter optimization* (HPO) algorithms [Bischl et al., 2023] aim to efficiently traverse a predefined search space to find good configurations quickly and avoid unpromising regions. Generally, HPO is framed as optimizing an expensive black-box function since the true functional form of the objective is commonly unknown, and the evaluation of hyperparameter configurations is costly, as it requires training ML models several times. Bayesian optimization (BO) methods have proven to be well-suited for HPO since they are sample efficient and converge on good configurations quickly. Typically, BO algorithms approximate the black-box objective with a surrogate model based on observations made during optimization and use the surrogate in an acquisition function to select the next candidate configuration, balancing exploration of the search space and exploitation of the surrogate [Shahriari et al., 2016, Wang et al., 2022].

   Although the recent advancements in HPO could facilitate the design and optimization of ML models for non-experts, in most cases, hyperparameters are still tuned manually [Bouthillier and Varoquaux, 2020]. Given that many ML practitioners perform hyperparameter tuning purely based on their knowledge, experience, and intuition, integrating this valuable knowledge to guide HPO algorithms during optimization can substantially foster the search and mitigate its cost. Moreover, it makes HPO more flexible and interactive, bringing it closer to the recently envisioned goal of human-centered AutoML [Lindauer et al., 2024]. For example, in Fig. 1 (Left), three hyperparameters of a CNN are optimized (depth multiplier $N$, width multiplier $W$, and resolution $R$). During an

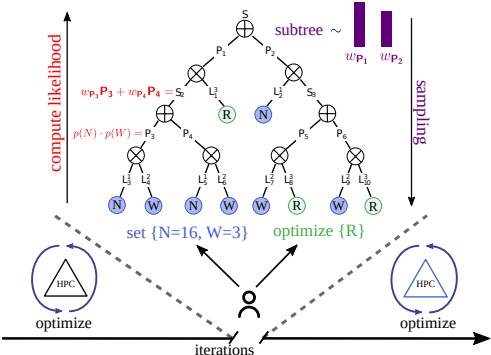 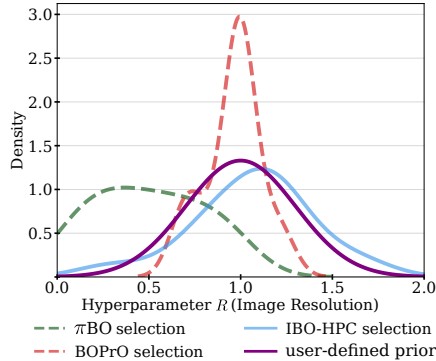

Figure 1: **Interactive Bayesian Hyperparameter Optimization**. (Left) We devise an interactive BO method by employing PCs as surrogates encoding a joint distribution over hyperparameters and evaluation scores (omitted for clarity). PCs allow users to directly condition the surrogate on their beliefs during tractable candidate generation, thereby reflecting user knowledge accurately. (Right) Accurately reflecting user beliefs is crucial for interactive HPO to fully leverage user knowledge. In contrast to $\pi$BO and BOPrO, IBO-HPC (our method) precisely reflects the user prior provided for hyperparameter $R$ (resolution). See App. A for details.

HPO run, a user might realize that values around $N = 16$ and $W = 3$ yield high-performing models. Hence, a user can guide the HPO algorithm with the obtained knowledge (here, $N = 16$ and $W = 3$) without restarting the optimization from scratch. This can considerably increase the convergence speed and quality of the final solution by focusing the optimization on remaining hyperparameters (here $R$, details in App. A). Recent works by Souza et al. [2021] and Hvarfner et al. [2022] allow users to infuse knowledge into a BO framework via user-defined prior distributions that are used to reshape the acquisition function according to user beliefs via multiplicative weighting. Although these approaches are valid and principled ways to guide an HPO task, their weighting schemes might not reflect user knowledge accurately. For example, in Fig. 1 (Right), the configurations selected by both BOPrO [Souza et al., 2021] and $\pi$BO [Hvarfner et al., 2022] during the first 20 iterations of optimization remarkably deviate from the given user prior (see App. A for details). This happens because acquisition functions are essentially black-boxes, often non-convex and hard to optimize [Wilson et al., 2018]. Consequently, it is hard to anticipate the influence of user beliefs on the actual behavior of the BO algorithm. Note that although trivial incorporation of user beliefs, e.g., by setting certain hyperparameters when optimizing the acquisition function, can be accurate, the acquisition function remains a black-box and the influence of priors is hard to anticipate.

To integrate user feedback in HPO more reliably and accurately, we introduce INTERACTIVE BAYESIAN OPTIMIZATION VIA HYPERPARAMETER PROBABILISTIC CIRCUITS (IBO-HPC). This novel BO method relies on probabilistic circuits (PCs) [Choi et al., 2020] as a surrogate model. PCs encode joint distributions over a set of random variables and come with exact and tractable marginalization, conditioning, and sampling. We derive an acquisition function-free, purely data-driven selection policy that suggests new configuration candidates by leveraging PCs' tractable conditional sampling mechanism. User beliefs can be provided at any time and are reflected accurately by directly conditioning the PC on them during candidate selection (see Fig. 1).

**Our Contributions**: **(1)** We introduce a novel BO method (IBO-HPC) that does not require any inner-loop optimization of an acquisition function and enables a direct, accurate, flexible, and targeted incorporation of user knowledge into BO. **(2)** We formally define a general notion of interactive policy in BO, and show that IBO-HPC conforms to this notion and is guaranteed to reflect user knowledge accurately during optimization. **(3)** We analyze the convergence of IBO-HPC and show that it converges proportionally to the expected improvement. **(4)** We provide an extensive empirical evaluation of IBO-HPC showing that it is competitive with strong HPO baselines on standard HPO tasks and outperforms strong interactive baselines in interactive HPO.

## 2 Background & Related Work

This section briefly introduces hyperparameter optimization (HPO) as well as relevant related work, including (interactive) Bayesian optimization (BO) and probabilistic circuits (PCs). HPO is formally defined as follows [Kohavi and John, 1995, Hutter et al., 2019].

**Definition 1** (Hyperparameter optimization (HPO)). *Given hyperparameters $\mathcal{H} = \{H_1, \ldots, H_n\}$ with associated domains $\mathbf{H}_1, \ldots, \mathbf{H}_n$, we define a search space $\Theta = \mathbf{H}_1 \times \cdots \times \mathbf{H}_n$. For a given black-box evaluation function $f : \Theta \rightarrow \mathbb{R}$, hyperparameter optimization aims to solve $\theta^* = \arg\min_{\theta \in \Theta} f(\theta)$.*

BO has effectively solved many practically relevant HPO tasks and will be introduced next.

**Interactive Bayesian Optimization**. BO aims to optimize a black-box objective function $f : \Theta \rightarrow \mathbb{R}$ which is costly to evaluate, i.e., to find the input $\theta^* = \arg\min_{\theta \in \Theta} f(\theta)$ [Shahriari et al., 2016]. BO typically leverages two key ingredients, a probabilistic surrogate model and a selection policy determining the next $\theta'$ to be evaluated, and uses them as follows in each iteration: Given a set $\mathcal{D}_n$ of observations that correspond to the configurations with associated evaluations $(\theta_j, f(\theta_j))_{j=1\ldots n}$, the probabilistic surrogate $s$ aims to approximate $f$ as closely as possible. Common choices for surrogate models are Gaussian processes (GPs) [Rasmussen and Williams, 2006] or random forests (RFs) [Breiman, 2001]. The selection policy uses $s$ to select the next $\theta' \in \Theta$ s.t. it achieves a good exploration–exploitation trade-off. Prominent selection policies optimize an acquisition function $a_s : \Theta \rightarrow \mathbb{R}$, such as expected improvement (EI) [Jones et al., 1998] that estimates the utility of an evaluation at an arbitrary point $\theta \in \Theta$ under a surrogate $s$, or perform Thompson sampling [Wang et al., 2022]. Various approaches to BO with different surrogates and acquisition functions have been proposed [Mockus, 1975, Hutter et al., 2011, Snoek et al., 2012, Shahriari et al., 2016].

To increase the efficiency of HPO, Hvarfner et al. [2022] and Souza et al. [2021] allow users to provide prior beliefs via prior distributions over the search space. The prior is used to multiplicatively re-weight the acquisition function's values according to the prior when selecting new configurations, thus, favoring configurations with a high likelihood in the prior. Mallik et al. [2023] propose a similar mechanism to incorporate user knowledge in multi-fidelity optimization. As illustrated in Sec. 1, these approaches present several drawbacks in reflecting user knowledge well when selecting new configurations. Moreover, additional constraints such as invertible priors [Ramachandran et al., 2020] or a specific acquisition function [Souza et al., 2021] are often required.

**Probabilistic Circuits (PCs)**. Probabilistic circuits [Choi et al., 2020] are computational graphs that compactly represent multivariate distributions. PCs provide exact inference for a wide range of probabilistic queries in a tractable fashion and can (conditionally) generate new samples. We leverage these properties to design a policy that accurately adheres to given user beliefs. More formally, a PC is a computational graph encoding a distribution over a set of random variables $\mathbf{X}$. It is defined as a tuple $(\mathcal{G}, \phi)$ where $\mathcal{G} = (V, E)$ is a rooted, directed acyclic graph and $\phi : V \rightarrow 2^{\mathbf{X}}$ is the *scope* function assigning a subset of random variables to each node in $\mathcal{G}$. For each internal node N of $\mathcal{G}$, the scope is defined as the union of scopes of its children, i.e., $\phi(\mathsf{N}) = \cup_{\mathsf{N}' \in \mathrm{ch}(\mathsf{N})} \phi(\mathsf{N}')$. Each leaf node L computes a distribution/density over its scope $\phi(\mathsf{L})$. All internal nodes of $\mathcal{G}$ are either a (weighted) sum node S or a product node P where each sum node computes a convex combination of its children, i.e., $\mathsf{S} = \sum_{\mathsf{N} \in \mathrm{ch}(\mathsf{S})} w_{\mathsf{S},\mathsf{N}} \mathsf{N}$, and each product node computes a product of its children, i.e., $\mathsf{P} = \prod_{\mathsf{N} \in \mathrm{ch}(\mathsf{P})} \mathsf{N}$. We employ *smooth* and *decomposable* PCs (see App. C for details), thus, our method can exploit tractable inference, sampling, and conditioning of valid and efficient PCs. For a more detailed description of PCs, refer to App. C; for an overview, see Fig. 1 (Left). We jointly model the hyperparameters and evaluation scores with PCs to guide the optimization towards promising solutions. Given the hybrid (discrete and continuous) nature of hyperparameter search spaces, IBO-HPC relies on mixed sum-product networks (MSPNs) [Molina et al., 2018], a decomposable and smooth PC with piecewise polynomial leaves, tailored to model hybrid domains.

## 3 Interactive Hyperparameter Optimization

We now introduce a formal notion of interactivity in BO to foster a more theoretically grounded approach to interactive BO. Then, we present IBO-HPC and our novel *feedback-adhering interactive* selection policy which reflects user beliefs accurately and does not require inner-loop optimization of an acquisition function. Lastly, we conduct a theoretical analysis of IBO-HPC.

### 3.1 Interactivity in Bayesian HPO

An interactive BO method should be capable of incorporating, at any time, the knowledge provided by users, and the selection policy should accurately reflect the provided user belief. Consequently, we formalize the concept of an interactive selection policy that adheres to these requirements and is compatible with a broad set of the possible types of user knowledge $\mathcal{K}$ (see App. B for details).

**Definition 2** (Feedback-Adhering Interactive Policy). *Given user knowledge $\mathcal{K} \in \mathcal{K}$ and surrogate $s$, an interactive policy $g_s$ is a function $g_s : \Theta \times \mathcal{K} \to \mathcal{P}(\Theta)$ mapping from the search space $\Theta$ to the set of all distributions $\mathcal{P}(\Theta)$ over $\Theta$. $g_s$ is called efficacious if $g_s(\Theta, \mathcal{K}) \neq g_s(\Theta, \emptyset)$ where $\emptyset$ indicates that $g_s$ is applied without user knowledge. If further $\mathcal{K}$ is provided as a distribution $q(\hat{\mathcal{H}})$ over $\hat{\mathcal{H}} \subset \mathcal{H}$, we call $g_s$ feedback-adhering if it is efficacious and $\int_{\mathcal{H}'} g_s(\Theta, \mathcal{K}) = q(\hat{\mathcal{H}})$ holds where $\mathcal{H}' = \mathcal{H} \setminus \hat{\mathcal{H}}$, i.e., the distribution over $\hat{\mathcal{H}}$ induced by the selection policy equals the prior $q(\hat{\mathcal{H}})$ in the next iteration.*

In Def. 2, being efficacious ensures that the user knowledge provided to $g_s$ has an effect on the sampling policy. The feedback-adhering condition ensures that in the *first* iteration, after a user provides a distribution over a subset of hyperparameters, the values sampled for the specified hyperparameters follow the distribution $q$ given by the user. Note that user knowledge could also be misleading, thus, Def. 2 does not guarantee user knowledge to have exclusively positive effects. We now introduce IBO-HPC that adheres to Def. 2.

### 3.2 Interactive Bayesian Optimization with Hyperparameter Probabilistic Circuits

To design an interactive BO method that reflects user beliefs accurately and enables flexible interactions with the optimization procedure by providing an arbitrary amount of knowledge about hyperparameters at any iteration, we construct a policy that leverages PCs as surrogates. We now describe our method in detail (see Algo. 1).

**Method.** Since PCs are density estimators, we start off by sampling $J$ hyperparameter configurations from a prior distribution $u$ and evaluate them by querying the objective function $f$ (**Line 2-3**). The function $f$ yields the (noisy) performance score of the sampled configuration $\theta$. After evaluating each sampled $\theta$ we obtain a set $\mathcal{D}$ of pairs $(\theta, f(\theta))$. We fit a PC $s(\mathcal{H}, F)$ that models the observations $\mathcal{D}$ as a joint distribution over hyperparameters $\mathcal{H}$ and evaluation score $F$ by maximizing the likelihood of $\mathcal{D}$ (**Line 5**). Both hyperparameters $\mathcal{H}$ and evaluation score $F$ are treated as random variables and assumed to follow a ground truth distribution $p(\mathcal{H}, F)$ that is approximated by $s$. Next, a configuration $\theta$ is selected by our *feedback-adhering interactive policy* that gets evaluated. Our policy exploits the flexible and exact inference of PCs to derive arbitrary conditional distributions according to the partial evidence at hand [Peharz et al., 2015]. We target the configurations that are likely to achieve a better evaluation score. Thus, a posterior distribution over the hyperparameter space is derived by conditioning on the best score $f^* = \max_f \mathcal{D}$ observed so far alongside with (optional) user knowledge $\mathcal{K}$. For now, $\mathcal{K}$ is assumed to be given in the form of conditions such as $\hat{\mathcal{H}} = \hat{\theta}$ where $\hat{\mathcal{H}} \subset \mathcal{H}$ is a subset of hyperparameters being set to $\hat{\theta}$. Using Bayes rule, tractable marginal inference, and sampling of PCs, we obtain the conditional distribution and use it to sample a new configuration from promising regions in the search space. Setting $\mathcal{H}' = \mathcal{H} \setminus \hat{\mathcal{H}}$, we perform sampling by:

$$s(\mathcal{H}'|\hat{\mathcal{H}}, F) = \frac{s(\mathcal{H}, F)}{\int_{\mathcal{H}'} s(\mathcal{H}, F)} \text{, then sample} \quad \theta \sim s(\mathcal{H}'|\hat{\mathcal{H}}, F = f^*). \tag{1}$$

App. C details how to marginalize, condition, and sample in PCs. Since users might be uncertain about hyperparameter values, defining a prior $q(\hat{\mathcal{H}})$ over $\hat{\mathcal{H}}$ might be more reasonable than setting a fixed value for certain hyperparameters. The prior $q(\hat{\mathcal{H}})$ is interpreted as a distribution over conditions of the form $\hat{\mathcal{H}} = \hat{\theta}$ where $\hat{\theta} \sim q(\hat{\mathcal{H}})$. This weights the distribution from Eq. 1 with the user prior. We then obtain $s(\mathcal{H}'|\hat{\mathcal{H}}, F = f^*) \cdot q(\hat{\mathcal{H}})$. Conditioning $s$ on user knowledge ensures that the provided user knowledge is precisely reflected in the next candidates; also, conditioning $s$ on $f^*$ ensures that only promising configurations are likely to be selected, allowing us to select new candidate configurations by mere sampling, thus, avoiding an inner loop optimization of an acquisition function. Since user intuitions can be wrong, we allow IBO-HPC to recover from misleading user knowledge $\mathcal{K}$ by deciding whether or not to use the provided

**Algorithm 1 IBO-HPC**

1: **Input:** Search space $\Theta$ over $\mathcal{H}$, initial prior distribution $u(\mathcal{H})$, objective $f : \Theta \rightarrow \mathbb{R}$, user prior $q(\hat{\mathcal{H}})$ (can be provided at any time), decay $\gamma$
2: Sample $J$ configurations $\theta \sim u(\mathcal{H})$
3: $\mathcal{D} \leftarrow \{(\theta_i, f(\theta_i)\}$ for $i \in \{1, ..., J\}$
4: **while** not converged **do**
5:     Fit PC $s$ on $\mathcal{D}$ every $L$-th iteration
6:     Set $f^* \leftarrow \max_f \mathcal{D}$ and $b \sim \text{Ber}(\rho)$
7:     **if** prior $q(\hat{\mathcal{H}})$ is given and $b = 1$ **then**
8:         Sample $N$ conditions $\theta \sim q(\hat{\mathcal{H}})$
9:         $C \leftarrow \emptyset$
10:        **for** condition $\theta_i$ in $\theta$ **do**
11:           Sample $\theta'_{1,...,B} \sim s(\mathcal{H}'|\hat{\mathcal{H}}, f^*)$
12:           $\theta^*_i \leftarrow \arg\max_{\theta' \in \theta'_{1,...,B}} s(\theta'|f^*)$
13:           $C \leftarrow C \cup \theta^*_i$
14:        $\theta^* \sim \mathcal{U}(C)$
15:     **else**
16:        $\theta^* \sim s(\mathcal{H}|f^*)$
17:     set $\mathcal{D} \leftarrow \mathcal{D} \cup \{(\theta', f(\theta'))\}$ and $\rho \leftarrow \gamma \cdot \rho$

$\mathcal{K}$ based on a Bernoulli distribution with success probability $\rho$. To achieve this, we gradually decrease the likelihood of using $\mathcal{K}$ in each iteration after $\mathcal{K}$ is supplied via a decay factor $\gamma$. For user knowledge provided at iteration $T$, the distribution over configurations after $T + t$ iterations reads:

$$\gamma^t \rho \cdot s(\mathcal{H}'|\hat{\mathcal{H}}, F = f^*) \cdot q(\hat{\mathcal{H}}) + (1 - \gamma^t \rho) \cdot s(\mathcal{H}|F = f^*). \tag{2}$$

Note that fusing the prior $q(\hat{\mathcal{H}})$ with the PC to allow exact inference and conditioning is non-trivial in general since the prior is defined over an arbitrary subset, and no further assumptions about the prior are made (except efficient sampling from $q(\hat{\mathcal{H}})$). Thus, we approximate the first mixture component of Eq. 2 by sampling $N$ times from $q(\hat{\mathcal{H}})$ and use Eq. 1 to obtain $N$ conditional distributions respecting the user prior $q(\hat{\mathcal{H}})$. We sample $B$ configurations from each conditional to ensure a certain amount of exploration in each iteration. For each conditional, the configuration maximizing the likelihood $s(\mathcal{H}|F = f^*)$ is selected to reduce the candidate set to configurations likely to achieve a high evaluation score. This leaves us with $N$ configurations from which we sample uniformly to select the configuration evaluated next **(Line 6-16)**. We found that setting $B = 1$ works surprisingly well. A discussion about the quality of our approximation and an analysis of the exploration-exploitation trade-off are given in App. D.4 and E.7 respectively. The surrogate is kept fixed for $L$ optimization rounds before retraining it. This fosters exploration by leveraging uncertainty encoded in the (conditional) distribution of the surrogate. An iteration is concluded by updating the set of evaluations $\mathcal{D}$ that can be presented to the users **(Line 17)**. The algorithm runs until convergence or another condition for termination, e.g., a time budget limit is encountered.

**Remark 1.** *Although similar, our sampling policy differs from Thompson Sampling (TS): TS samples function values from the posterior and selects the next configuration based on the sampled function's maximum. Instead, we use the maximum obtained so far to condition the surrogate and sample from it (via conditional sampling) the next configuration.*

### 3.3 Theoretical Analysis

Let us now analyze the theoretical properties of IBO-HPC. We start by showing that the presented selection policy is feedback-adhering interactive according to Def. 2.

**Proposition 1** (IBO-HPC Policy is feedback-adhering interactive). *Assume a search space $\Theta$ over hyperparameters $\mathcal{H}$, a PC $s$, user knowledge $\mathcal{K} \in \mathcal{K}$ in form of a prior $q$ over $\hat{\mathcal{H}} \subset \mathcal{H}$ s.t. the marginal over $\hat{\mathcal{H}}$ of $s$ conditioned on $f^*$ is different than $q(\hat{\mathcal{H}})$, i.e., $\int_{\mathcal{H}'} s(\hat{\mathcal{H}}|F = f^*) \neq q(\hat{\mathcal{H}})$ with $\mathcal{H}' = \mathcal{H} \setminus \hat{\mathcal{H}}$. Then, the selection policy of IBO-HPC is feedback-adhering interactive. See App. D.1 for the proof.*

Besides being feedback-adhering, IBO-HPC is a global optimizer for black-box optimization.

**Proposition 2** (IBO-HPC is a global optimizer). *IBO-HPC minimizes simple regret, which is defined as $r = f(\theta) - f(\theta^*)$ for a hyperparameter configuration $\theta \in \Theta$ and global optimum $\theta^*$. A proof is given in App. D.2.*

Next, we analyze IBO-HPC's convergence behavior in each iteration by deriving our algorithm's expected improvement (EI) with a surrogate PC $s$.

**Proposition 3** (Convergence of IBO-HPC). *Assume a differentiable $L$-Lipschitz continuous function $f : \mathbb{R}^d \to \mathbb{R}$ and a (local) optimum $\theta^* \in \mathbb{R}^d$. Assume $f$ is locally convex within a ball $B_r(\theta^*) = \{\theta \in \mathbb{R}^d : ||\theta - \theta^*|| < r\}$. Furthermore, assume we have a dataset $\mathcal{D} = \{(\theta_1, y_1), \ldots, (\theta_n, y_n)\}$ where $\theta_i \in B_r$, $y_i = f(\theta_i)$ and $s$ is a decomposable, smooth PC over $\mathcal{H} \cup \{F\}$ where the support of $\mathcal{H} = B_r$ and the support of $F = \mathbb{R}$. Assume $s$ locally maximizes the likelihood over $\mathcal{D}$ and that all leaves are Gaussians. Then, the lower bound of the convergence rate of IBO-HPC is given by the expected improvement (EI) in each iteration:*

$$\sum_{i=1}^{\tau_s} w_i \cdot \Big( \prod_{j=1}^{d} erf\Big(\frac{\theta_{tj}^* - \mu_{ij}}{\Sigma_{i_{jj}}\sqrt{2}}\Big) - \prod_{j=1}^{d} erf\Big(\frac{\theta_j^* - \mu_{ij}}{\Sigma_{i_{jj}}\sqrt{2}}\Big) + L\epsilon_i \Big). \tag{3}$$

*Here, $\tau_s$ is the number of induced trees of $s$ (see Def. 4 in App. D.3), $\epsilon_i = ||\mu_i + \alpha_i \cdot diag(\Sigma_i) - \theta^*||$, each $\mu_i$ is the mean vector of a $d$-dimensional multivariate Gaussian defined by the $i$-th induced tree, $\Sigma_i$ is the corresponding correlation matrix and $\theta_t^*$ is the best performing configuration until iteration $t$. A proof is given in App. D.3.*

Intuitively, the convergence in each iteration is determined by (1) the probability of sampling a configuration closer to $\theta^*$ and (2) expected distance to move to the optimum $\theta^*$ if (1) occurs. (1) is lower bounded by the probability mass between the mixture means and the best obtained configuration $\theta_t^*$ at iteration $t$, and the probability mass between mixture means and the optimum $\theta^*$. (2) is lower bounded for each mixture component by $\epsilon_i$ and the Lipschitz constant $L$.

## 4 Experimental Evaluation

We now provide an extensive empirical evaluation of IBO-HPC and aim to answer the following research questions: **(Q1)** Can IBO-HPC compete with prominent HPO methods? **(Q2)** How does the performance of IBO-HPC, provided with user knowledge at various points during optimization, compare to existing approaches incorporating user knowledge ex ante? **(Q3)** Is IBO-HPC capable of reliably recovering from misleading user interactions?

**Experimental Setup.** We compare IBO-HPC against eight diverse competitors: local search (LS) [White et al., 2020], BO with random forest (RF) [Head et al.], BO with tree-parzen estimators (TPE) [Akiba et al., 2019], and SMAC [Hutter et al., 2011, Lindauer et al., 2022] as HPO methods that do not permit user interactions. As baselines allowing for ex ante incorporation of user interactions, we employ random search (RS) [Bergstra and Bengio, 2012] with user priors, BOPrO [Souza et al., 2021], $\pi$BO [Hvarfner et al., 2022], and Priorband [Mallik et al., 2023]. Since Priorband is a multi-fidelity method, we reserve the number of epochs as the fidelity in each benchmark. For all single-fidelity methods, we set it to the highest possible value. For our evaluation,

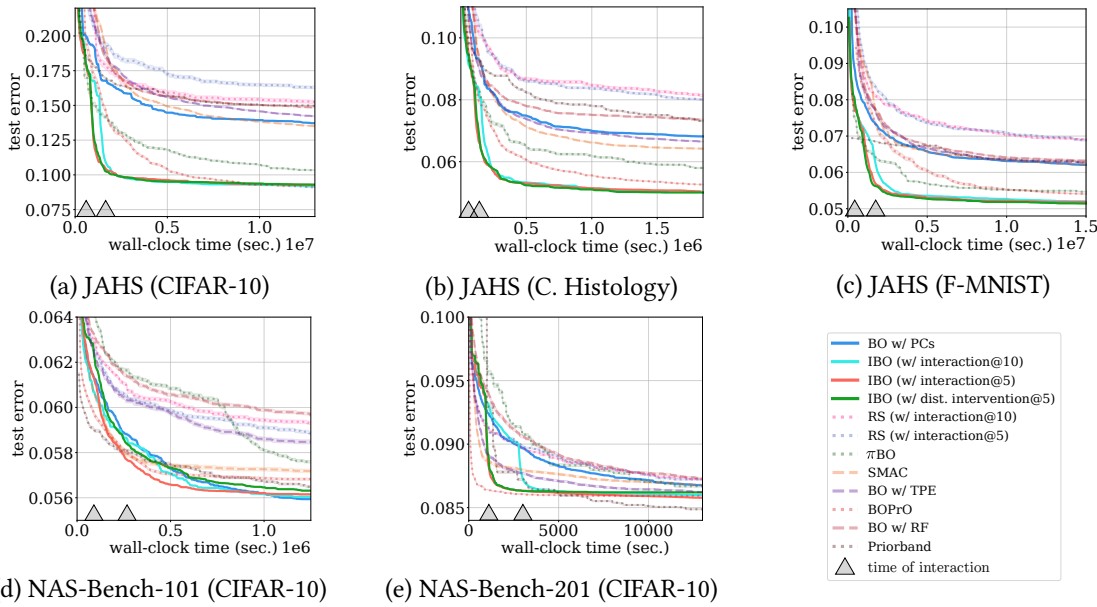

Figure 2: **IBO-HPC outperforms state of the art**. For 5/5 tasks across three challenging benchmarks, IBO-HPC is competitive with strong baselines when no user knowledge is provided. When beneficial user beliefs are provided (△), after 5 (—) or after 10 iterations (—), it outperforms all competitors w.r.t. convergence and/or solution quality on 4/5 tasks. Best viewed in color.

we employ six real-world benchmarks: NAS-Bench-101 [Ying et al., 2019], NAS-Bench-201 [Dong and Yang, 2020], JAHS [Bansal et al., 2022], HPO-B [Pineda-Arango et al., 2021], PD1 [Wang et al., 2024], and FCNet [Klein and Hutter, 2019]. See App. E.1 for an overview. From these benchmarks, we use 10 diverse tasks and 7 different search spaces covering continuous, discrete, and mixed spaces. Each of the 10 tasks considers either HPO, NAS or both. The tasks cover classification and regression tasks on tabular and image data (see App. E.1 for details). All algorithms optimize the validation accuracy. We report the mean test error against computational cost and provide the standard error to quantify uncertainty. All algorithms are run with 500 seeds for 200 iterations (50 seeds with 100 iterations for HPO-B, PD1, and FCNet) and were initialized with 5 random samples. The computational costs are reported as the accumulated wall-clock time of training and evaluation of each sampled configuration, provided by the benchmarks (App. E.2, E.8, and E.9 for more details).

**User Interactions**. We follow Souza et al. [2021] and Hvarfner et al. [2022] and define beneficial and misleading user beliefs. To define beneficial and misleading user knowledge/priors for each benchmark and the corresponding search space, we randomly sample $10k$ configurations and keep the best/worst performing ones, denoted as $\theta^+$ and $\theta^-$, respectively. To demonstrate that beneficial user priors **over only a few hyperparameters** are enough to improve the performance of IBO-HPC remarkably, we define beneficial interactions by selecting a small subset of hyperparameters $\hat{\mathcal{H}} \subset \mathcal{H}$. Then, we define a prior over each $H \in \hat{\mathcal{H}}$ favoring the value of $H$ given in $\theta^+$. For misleading interaction, $\hat{\mathcal{H}}$ is chosen to be large to demonstrate that IBO-HPC recovers even if a large amount of misleading information is provided. A prior is then defined s.t. values from $\theta^-[H]$ are favored. Since users might want to specify a concrete value for certain hyperparameters instead of defining a distribution, we also conduct experiments with point masses as user priors. We discuss interaction design in App. E.3.

## 4.1 (Q1) IBO-HPC is Competitive in HPO & NAS

To demonstrate the effectiveness of IBO-HPC, we ran IBO-HPC on all tasks without user interaction. We compared its performance against three strong BO baselines and LS (for LS, see Fig. 7). Fig. 2

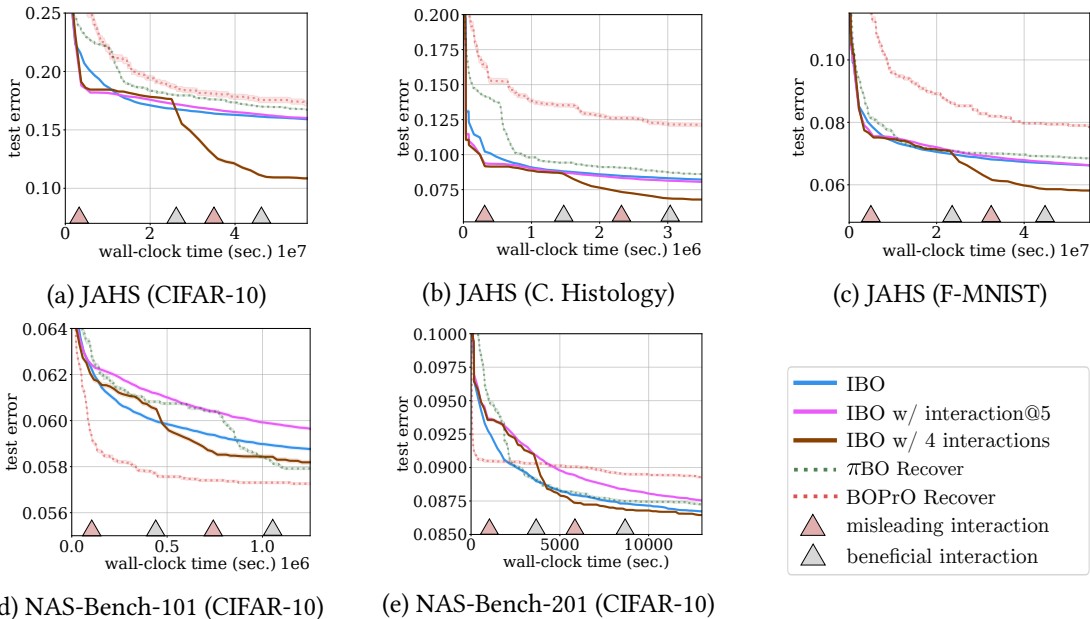

(a) JAHS (CIFAR-10)  (b) JAHS (C. Histology)  (c) JAHS (F-MNIST)

(d) NAS-Bench-101 (CIFAR-10)  (e) NAS-Bench-201 (CIFAR-10)

Figure 3: **IBO-HPC recovers from misleading interactions**. IBO-HPC automatically recovers from misleading feedback (——) provided as point values at the 5th iteration of the search (1st △). Also, when providing harmful and beneficial beliefs alternatively (△/△), IBO-HPC (——) catches up with or outperforms $\pi$BO (····) and BOPrO (····) in 4/5 cases. Best viewed in color.

and App. E.4 show that the performance of IBO-HPC without user interaction is competitive to or outperforms BO baselines on all selected benchmarks. These results show that IBO-HPC performs well in complex and realistic settings. Also, it underlines that HPCs accurately capture characteristics of the objective function and that our sampling-based selection policy reliably identifies good configurations. Besides the quality of the final solution, we also observe that IBO-HPC converges at rates similar to those of the baselines. We thus answer **(Q1)** affirmatively, since IBO-HPC is competitive with existing strong BO baselines without user interaction.

## 4.2 (Q2, Q3) IBO-HPC is Interactive and Resilient

We now demonstrate that IBO-HPC successfully handles point values and distributions as user knowledge, analyze the benefits of user knowledge w.r.t. convergence speed, and demonstrate IBO-HPC's recovery from misleading beliefs. Details about the different beneficial and misleading user beliefs on hyperparameters are described in App. E.3.

**Beneficial Interactions**. Fig. 2 and 7 show a clear positive effect of providing beneficial user beliefs (distribution or fixed values) to IBO-HPC across all tasks. This holds for very early interactions (after 5 iterations; —— and ——) and later interactions (after 10 iterations; ——). Remarkably, we observed a clear benefit in terms of convergence speed *and* improvement in solution quality, especially for more complex search spaces. Considering the case in which users provide knowledge, IBO-HPC outperforms $\pi$BO, Priorband, and BOPrO in 4/5 cases w.r.t. convergence speed and/or final performance (see App. E.5 for significance test). The results demonstrate that IBO-HPC's selection policy accurately represents the given user beliefs. Also, it shows that the selection policy effectively leverages information encoded in user priors *and* the surrogate since beneficial feedback provides decisive improvements, and then the optimization keeps improving. We found similar results on HPO-B, PD1, and FCNet (see App. E.4).

**Recovery and Multiple Interactions.** User beliefs could also mislead the optimization, thus, an interactive HPO algorithm should be able to recover from misleading interactions and allow users to correct their initial beliefs. We demonstrate the recovery mechanism of IBO-HPC by deliberately providing IBO-HPC with known sub-optimal values for a large subset of hyperparameters to ensure a significant negative effect on the optimization process (see App. E for details). We allowed a budget of $2k$ iterations for each algorithm to test the long-term effects of negative/multiple interactions. Fig. 3 shows that IBO-HPC (—) recovers similarly well or better than $\pi$BO and BOPrO from misleading interactions. In most cases, IBO-HPC catches up with standard HPO competitor methods (having no good/bad interactions). This confirms that

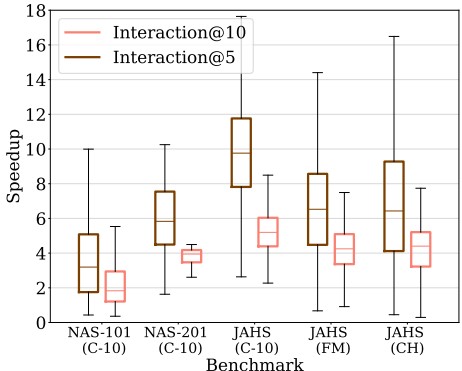

Figure 4: **IBO-HPC achieves considerable runtime improvement** with beneficial interactions **(2-10× faster)**.

IBO-HPC's recovery mechanism works reliably and that misleading user beliefs do not deteriorate IBO-HPC's performance in the long run. Users might revise their beliefs when no improvement is obtained. In extreme cases, users could alternate between beneficial and misleading interactions. Therefore, we first provided IBO-HPC with the same misleading beliefs as before, after 5 iterations, followed by an alternation of beneficial and harmful beliefs every 10 iterations. As expected, the misleading interactions decelerate IBO-HPC and trigger the recovery mechanism. Upon receiving beneficial knowledge, IBO-HPC catches up with or outperforms the baselines, confirming that IBO-HPC leverages valuable feedback in critical conditions (see Fig. 3 (—) and App. E.4).

**Speed-up.** Both the runtime of the optimization loop (fitting surrogate and suggesting the next configuration) and convergence speed are crucial for efficient HPO. Therefore, we analyze the increase in convergence speed when valuable user knowledge is provided to IBO-HPC as a distribution and the average runtime of IBO-HPC's optimization loop. To assess the speed-up of IBO-HPC due to beneficial user knowledge, we run IBO-HPC without user interaction and obtain the wall-clock time needed for the best evaluation result (denoted as $t_w$). Then, we run IBO-HPC with beneficial user knowledge and measure the estimated wall-clock time until IBO-HPC finds an equally good or better configuration (denoted as $t_i$). Fig. 4 reports the relative performance speedup $\frac{t_w}{t_i}$ for all 500 runs. A median speed-up of 2-10× with beneficial user interactions clearly demonstrates IBO-HPC's improvement in convergence speed while saving resources. Since SMAC is the best of our baselines and performs similarly to IBO-HPC when no user knowledge is provided, we compare the efficiency of the selection policies of SMAC and IBO-HPC, averaging over 20 runs (Fig. 14b in App. E.6). IBO-HPC is considerably faster than SMAC in 4/5 cases, especially in larger search spaces. These results (and those from **(Q1)**) demonstrate that our selection policy, leveraging conditional sampling of PCs, not only matches or surpasses the performance of BO methods requiring inner-loop optimization, but also significantly improves efficiency. Given IBO-HPC's remarkable speed-ups and the reliable recovery mechanism, we can answer **(Q2)** and **(Q3)** positively.

## 5 Conclusion

We introduced a novel definition of interactive BO policy and IBO-HPC, an interactive BO method that leverages the flexible inference of probabilistic circuits to accurately and flexibly incorporate user beliefs. Without user knowledge, IBO-HPC is competitive with strong baselines and outperforms interactive competitors when knowledge is available. It reliably recovers from misleading user beliefs and converges significantly faster when provided with valuable user knowledge.

**Limitations & Future Work.** So far, IBO-HPC only allows users to provide external knowledge about a given HPO task but does not provide a way to leverage information from previous HPO

runs performed on different tasks. Therefore, a promising prospect for future research is the usage of PCs to enable hyperparameter transfer learning, thus, incorporating both former HPO runs and user knowledge to make HPO more efficient. See App. F for a more detailed discussion. Moreover, IBO-HPC can get stuck in local optima if the surrogate PC's leaves exhibit too low variance for a given task due to its sampling-based exploration. Although this can be tackled by setting a minimal variance or introducing a minimum variance schedule, this introduces new hyperparameters in IBO-HPC itself. These parameters must be set accordingly for each task by the user, which could be challenging for non-experts.

**Impact Statement**. After careful reflection, the authors have determined that this work presents no notable negative impacts to society or the environment.

## Acknowledgements

This work was supported by the National High-Performance Computing Project for Computational Engineering Sciences (NHR4CES) and the Federal Ministry of Education and Research (BMBF) Competence Center for AI and Labour ("KompAKI", FKZ 02L19C150). Furthermore, this work benefited from the cluster project "The Third Wave of AI".

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

## A  Motivation & Further Related Work

### A.1  Real-World Example

As highlighted by Lindauer et al. [2024], one crucial limitation of AutoML systems is the lack of incorporation of humans in the AutoML process. One crucial aspect of AutoML is HPO and NAS, where recent works aim to incorporate human knowledge into the optimization process. Reflecting user knowledge accurately is crucial for interactive HPO methods to fully benefit from human knowledge and improve trustworthiness. Existing weighting scheme-based methods like $\pi$BO and BOPrO fail to reflect user priors accurately in their selection policy, as seen in Fig. 5 (a). Here, we show a 1d-example of a Branin function with an optimum around $x = 0.5$. The user prior (in red) is placed at $x = 0.3$. Both $\pi$BO and BOPrO fail to select the next configuration from the high-density region of the prior; thus, the user prior is not incorporated in the selection process as a user would expect. We followed the recommendation of Hvarfner et al. [2022] and set $\beta = \frac{T}{10}$ where we ran $\pi$BO for $T = 10$ iterations. The reason for this behavior is the fact that both $\pi$BO and BOPrO reshape the curvature of the acquisition function either directly ($\pi$BO) or indirectly via the surrogate (BOPrO). The objective function's curvature, the acquisition function's curvature, and how they behave when weighed against each other are unknown and not intuitively visualizable to the user (due to high dimensionality). Thus, it is non-trivial for users to define a prior that is strong enough to guarantee that $\pi$BO and BOPrO sample at desired regions but also weak enough to not fully govern the maximum of the acquisition function in early iterations. The latter allows $\pi$BO and BOPrO to leverage knowledge encoded in the surrogate model *and* the user knowledge at early iterations. Our method, IBO-HPC, solves this issue, which we demonstrate based on a real-world example (see below). This is achieved by circumventing the need of users to reshape an unknown acquisition function using a prior. Instead, users can provide their beliefs directly by conditioning the surrogate on their beliefs. Since the search space is static once defined and the surrogate model treats each dimension of the search space as a random variable, it is intuitive for users to define a prior s.t. the selection policy is guided to exactly the location where the prior was defined. In fact, a user prior defined over a certain hyperparameter can be interpreted as shifting the joint distribution over hyperparameters and evaluation scores (represented by our surrogate model) along the dimension corresponding to the hyperparameter the prior is defined over.

**Details on Fig. 5.** To demonstrate that IBO-HPC reflects user knowledge more accurately than $\pi$BO and BOPrO, we ran $\pi$BO, BOPrO – both of which leverage a weighting scheme to incorporate user priors –, and IBO-HPC for $T = 100$ iterations on the CIFAR-10 task of the JAHS benchmark [Bansal et al., 2022]. Following Hvarfner et al. [2022], we set the decay parameter of $\pi$BO to 10. We specified a Gaussian prior distribution with $\mu = 1$ and $\sigma = 0.3$ (Fig. 5 (b), purple) over the hyperparameter Resolution ($R$) that controls the down-/up-sampling rate of an image fed into a neural network. The rest of the hyperparameters for this specific task (i.e. the network architecture and all other hyperparameters; see App. D for details) were optimized by $\pi$BO, BOPrO and IBO-HPC without any user knowledge. All methods received the same user prior ($\pi$BO and BOPrO from the beginning of the optimization; IBO-HPC after 5 iterations). From the iteration the user prior was provided on, we then considered the values chosen for Resolution by $\pi$BO, BOPrO, and IBO-HPC for the next 20 iterations and estimated a density of selected values for $R$ (see Fig. 5 (b)). We chose 20 as the horizon under consideration because for higher $\beta$, the prior is weighted down later in $\pi$BO (see [Hvarfner et al., 2022], Alg. 1) and BOPrO (see [Souza et al., 2021] Eq. 4). In the JAHS setup with $T = 100$ and $\beta = 10$, the prior is weighted down after the 10th iteration in $\pi$BO and BOPrO. In the 20th iteration, $\pi$BO and BOPrO exponentially weigh down the prior with exponent 0.5. The density value of the mode of our prior is then $1.26^{0.5} \approx 1.12$. For IBO-HPC, we chose the decay $\gamma = 0.995$; hence, after 20 iterations, we get $1.26 \cdot \gamma^{20} \approx 1.14$ for the mode of the prior. Thus, we weigh down the prior by approximately the same factor in $\pi$BO, BOPro, and IBO-HPC, ensuring a fair comparison. We obtained that neither the choices for $R$ by $\pi$BO (green

dashed line) nor the choices of BOPrO (red dashed line) reflect the user prior as specified. While $\pi$BO's choices of RESOLUTION are biased towards smaller values, BOPrO does not reflect the user's uncertainty well in its choices of RESOLUTION. In contrast, IBO-HPC (blue solid line) precisely reflects the user prior as specified (up to random variations due to sampling).

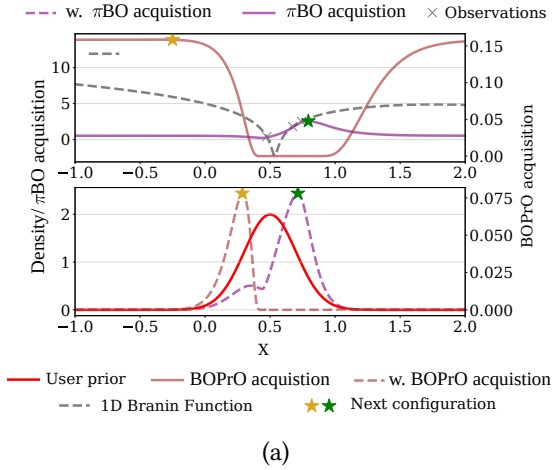 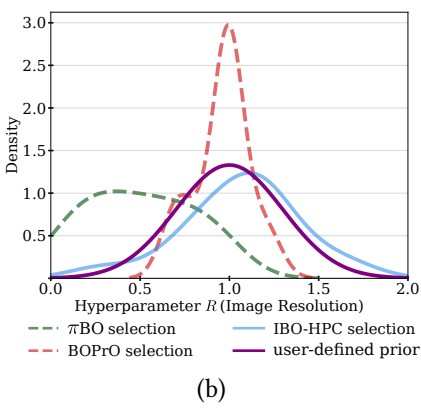

(a)

(b)

Figure 5: **IBO-HPC reflects user priors as specified**. In contrast to other weighting scheme based methods like $\pi$BO and BOPrO, IBO-HPC reflects the user prior as specified in its selection policy.

## A.2 Related Work: Hyperparameter Transfer Learning and Benchmarks

Hyperparameter Transfer Learning (HTL) uses information about former HPO runs, usually stored in logs, to increase the efficiency of subsequent optimization runs of similar HPO tasks. Prominent approaches perform HTL by projecting objective responses of all runs to a common response surface or by pruning the search space based on previous tasks [Yogatama and Mann, 2014, Wistuba et al., 2015, Perrone et al., 2018, Vanschoren, 2018, Salinas et al., 2020, Horváth et al., 2021].

Since HPO is costly and empirical results often depend on the exact definition of tasks, we use benchmarks for our empirical evaluation. Benchmarks foster the development, reproducibility, and fair comparison of HPO algorithms by defining a search space over hyperparameters and training all candidates to provide quantities like validation and test accuracy. Relevant examples covering hyperparameter optimization and neural architecture search are HPO-B [Pineda-Arango et al., 2021], NAS-Bench-101/201 [Ying et al., 2019, Dong and Yang, 2020] and JAHS [Bansal et al., 2022].

## B Details on User Knowledge

This section briefly discusses the theoretical details about the set of the possible relevant (user) knowledge $\mathcal{K}$. For a discussion on the user knowledge used in our experiments, refer to App. E.3.

In general, $\mathcal{K}$ refers to a set of possible objects a user can provide to guide the optimization process. In our case, all elements in $\mathcal{K}$ are assumed to be either in the form of user priors $q(\hat{\mathcal{H}})$ or assignments of hyperparameters to a certain value, i.e., $\hat{\mathcal{H}} = \hat{\boldsymbol{\theta}}$. Here, $\hat{\mathcal{H}}$ refers to a subset of hyperparameters that define the search space. However, other forms of user knowledge are possible. For example, users could also specify believes about possible correlations between hyperparameters or between hyperparameters and the evaluation score. We decided not to restrict $\mathcal{K}$ for our definition of feedback-adhering policies to generalize this notion to a broad set of types of user knowledge.

## C  Probabilistic Circuits

Since probabilistic circuits (PCs) are a key component of our method, we provide more details on these models in the following. Let us first start with a rigorous definition of PCs.

**Definition 3.** *A probabilistic cricuit (PC) is a computational graph encoding a distribution over a set of random variables* $\mathbf{X}$. *It is defined as a tuple* $(\mathcal{G}, \phi)$ *where* $\mathcal{G} = (V, E)$ *is a rooted, directed acyclic graph and* $\phi : V \to 2^{\mathbf{X}}$ *is the scope function assigning a subset of random variables to each node in* $\mathcal{G}$. *For each internal node* $\mathsf{N}$ *of* $\mathcal{G}$, *the scope is defined as the union of scopes of its children, i.e.* $\phi(\mathsf{N}) = \cup_{\mathsf{N}' \in \mathrm{ch}(\mathsf{N})}$. *Each leaf node* $\mathsf{L}$ *computes a distribution/density over its scope* $\phi(\mathsf{L})$. *All internal nodes of* $\mathcal{G}$ *are either a sum node* $\mathsf{S}$ *or a product node* $\mathsf{P}$ *where each sum node computes a convex combination of its children, i.e.,* $\mathsf{S} = \sum_{\mathsf{N} \in \mathrm{ch}(\mathsf{S})} w_{\mathsf{S},\mathsf{N}} \mathsf{N}$, *and each product computes a product of its children, i.e.,* $\mathsf{P} = \prod_{\mathsf{N} \in \mathrm{ch}(\mathsf{P})} \mathsf{N}$.

With this definition at hand, we describe the tractable key operations of PCs relevant to our method in more detail.

**Inference**. Inference in PCs is a bottom-up procedure. To compute the probability of given evidence $\mathbf{X} = \mathbf{x}$, the densities of the leaf nodes are evaluated first. This yields a density value for each leaf. The leaf densities are then propagated bottom-up by computing all product/sum nodes. Eventually, the root node holds the probability/density of $\mathbf{x}$. Note that typically, multiple leaf nodes correspond to the same random variable. Thus, if the children of a sum node have the same scope, we can interpret sum nodes as mixture models. Conversely, if the children of a product node have *non*-overlapping scopes, a product node can be interpreted as a product distribution of two (independent) random variables. We call these two properties smoothness and decomposability. More formally, *smoothness* means that for each sum node $\mathsf{S} \in V$ it holds that $\phi(\mathsf{N}) = \phi(\mathsf{N}')$ for $\mathsf{N}, \mathsf{N}' \in \mathrm{ch}(\mathsf{S})$. *Decomposability* means that for each product node $\mathsf{P} \in V$ it holds that $\phi(\mathsf{N}) \cap \phi(\mathsf{N}') = \emptyset$ for $\mathsf{N}, \mathsf{N}' \in \mathrm{ch}(\mathsf{P})$, $\mathsf{N} \neq \mathsf{N}'$. Hence, PCs can be interpreted as hierarchical mixture models.

**Marginalization**. Decomposability implies that marginalization is tractable in PCs and can be done in linear time of the circuit size. This is because integrals that can be rewritten by nesting single-dimensional integrals can be computed only in terms of leaf integrals, which are assumed to be tractable as they follow certain distributions (e.g., Gaussian). Computing such nested integrals only in terms of leaf integrals is possible because single-dimensional integrals commute with the sum operation and affect only a single child of product nodes. For more details on the computational implications of decomposability, refer to [Peharz et al., 2015].

Practically, there are two ways to marginalize certain variables from the scope of a PC. One approach is structure-preserving, and marginalization is achieved by setting all leaves corresponding to the set of random variables that are supposed to be marginalized to 1. The second approach constructs a new PC representing the marginal distribution, i.e. the structure of the PC is changed. The second approach is beneficial if samples should be drawn from the marginalized PC because the sampling procedure remains the same, i.e. the PC is adopted to obtain the marginal distribution, not vice versa.

**Conditioning**. Computing a conditional distribution $p(\mathbf{X}_1 | \mathbf{X}_2) = \frac{p(\mathbf{X})}{\int_{\mathbf{x}_2} p(\mathbf{X})}$ where $\mathbf{X}_1 \cup \mathbf{X}_2 = \mathbf{X}$ and $\mathbf{X}_1 \cap \mathbf{X}_2 = \emptyset$ is achieved by combining marginalization (denominator) and inference (numerator). Since inference is tractable for PCs in general and marginalization is tractable for decomposable PCs, conditioning is also tractable.

**Sampling**. Sampling in PCs is a top-down procedure and recursively samples a sub-tree, starting at the root. Each sum node $\mathsf{S}$ holds a parameter vector $\mathbf{w}$ s.t. $\sum_{i=0}^{|\mathrm{ch}(\mathsf{S})|} \mathbf{w}_i = 1$. Based on the distribution induced by $\mathbf{w}$, one of the children of $\mathsf{S}$ is sampled as a sub-tree. By decomposability, the scope of the children of a product node is non-overlapping; thus, sampling from a product node

corresponds to sampling from all its child nodes. If a leaf node is reached, a sample is obtained from the distribution at that leaf.

**Learning**. Learning PCs consists of two steps: Identify the structure of the PC and learn the parameters of the PC. A common approach to learning both the structure and parameters is LearnSPN [Gens and Domingos, 2013]. We employ LearnSPN to learn the PC after obtaining new data. The basic idea of LearnSPN is to split the data by alternating clustering (i.e., split the data along the sample dimension) and independence tests (i.e., split the data along the features dimension). In other words, the data matrix is split by rows (samples) and columns (features). Usually, rows are clustered when the independence test fails in splitting the features. Clusters correspond to sum nodes in the learned PC, while product nodes correspond to successfully passed independence tests (assessing that two subsets of features are statistically independent). The parameters (i.e., weights of sum nodes) are set proportional to the cluster sizes of clusters represented by the child nodes of a sum node. Leaf parameters are commonly defined via maximum likelihood estimation.

## D Proofs

In this section we provide the proof of Proposition 1 of the main paper.

### D.1 IBO-HPC's Policy is Feedback-Adhering Interactive

**Proposition 1 (IBO-HPC Policy is feedback-adhering interactive)**. Given a search space $\Theta$ over hyperparameters $\mathcal{H}$, an PC $s$, user knowledge $\mathcal{K} \in \mathcal{K}$ in form of a prior $q$ over $\hat{\mathcal{H}} \subset \mathcal{H}$ s.t. $\int_{\mathcal{H} \setminus \hat{\mathcal{H}}} s(\mathcal{H}|F = f^*) \neq q(\hat{\mathcal{H}})$, the selection policy of IBO-HPC is feedback-adhering interactive.

*Proof.* We have to show that the policy of IBO-HPC is feedback-adhering, i.e. it conforms with Def. 3: The distribution over the configuration space used to obtain new configurations is different if user knowledge is provided from the distribution used if no user knowledge is provided (policy is efficacious) and the provided user knowledge is represented during configuration selection as specified (feedback-adhering).

We first show that the selection policy of IBO-HPC is efficacious.

**IBO-HPC selection policy is efficacious**. Since the decay mechanism allowing IBO-HPC to recover from misleading knowledge can be treated as a constant in each iteration, it is enough if $s(\mathcal{H} \setminus \hat{\mathcal{H}}|\hat{\mathcal{H}} = \hat{\theta}, F = f^*) \cdot q(\hat{\mathcal{H}} = \hat{\theta}) \neq s(\mathcal{H} \setminus \hat{\mathcal{H}}|\hat{\mathcal{H}} = \emptyset, F = f^*) \cdot q(\hat{\mathcal{H}} = \emptyset)$ holds for any surrogate $s$ representing a joint distribution over search space $\mathcal{H}$ and prior $q$ over $\hat{\mathcal{H}} \subset \mathcal{H}$ to make the policy efficacious. Note that we assume that $\mathcal{K}$ is given in form of a prior $q(\hat{\mathcal{H}})$ over $\hat{\mathcal{H}}$ as before. Since $\emptyset \notin \hat{\mathcal{H}}$ is assumed, our policy ignores any prior if no user knowledge is provided. Thus, in this case, the policy samples from the distribution

$$s(\mathcal{H}|F = f^*) = s(\mathcal{H} \setminus \hat{\mathcal{H}}|\hat{\mathcal{H}}, F = f^*) \cdot \int_{\mathcal{H} \setminus \hat{\mathcal{H}}} s(\mathcal{H}|F = f^*) \tag{4}$$

Since $s(\mathcal{H} \setminus \hat{\mathcal{H}}|\hat{\mathcal{H}}, F = f^*)$ is the same, regardless of whether user knowledge is given or not, user knowledge will lead to a different distribution if $\int_{\mathcal{H} \setminus \hat{\mathcal{H}}} s(\mathcal{H}|F = f^*) \neq q(\hat{\mathcal{H}})$ holds. Since Prop. 1 demands that this is the case, our policy is efficacious according to Def. 2.

We can now proceed and show feedback adherence of the IBO-HPC selection policy.

**IBO-HPC selection policy is feedback-adhering**. The proof that our policy is feedback-adhering directly follows by design: If a user prior $q(\hat{\mathcal{H}})$ is given, Eq. 3 is approximated by sampling $N$ conditions $\theta'_{1,\dots,N} \sim q(\hat{\mathcal{H}})$ and computing $N$ conditionals $s(\mathcal{H} \setminus \hat{\mathcal{H}}|\hat{\mathcal{H}} = \theta'_1, F = f^*), \dots, s(\mathcal{H} \setminus \hat{\mathcal{H}}|\hat{\mathcal{H}} = \theta'_N, F = f^*)$. We can approximate $q(\hat{\mathcal{H}})$ arbitrarily close with $N \to \infty$. To select the next configuration, we sample $B$ configurations from each of the $N$ conditionals and select the configuration maximizing $s(\mathcal{H}|F = f^*)$ for each conditional. This leaves us with $N$ candidates. Note that at this point, the hyperparameters $\hat{\mathcal{H}}$ still follow $q(\hat{\mathcal{H}})$ with $N \to \infty$ as the conditions of

$s(\mathcal{H} \setminus \hat{\mathcal{H}} | \hat{\mathcal{H}} = \boldsymbol{\theta}'_1, F = f^*), \ldots, s(\mathcal{H} \setminus \hat{\mathcal{H}} | \hat{\mathcal{H}} = \boldsymbol{\theta}'_N, F = f^*)$ remain fixed and only hyperparameters of the set $\mathcal{H} \setminus \hat{\mathcal{H}}$ can vary/are sampled. Thus, maximizing the likelihood $s(\mathcal{H}|F = f^*)$ is only done w.r.t. hyperparameters in $\mathcal{H} \setminus \hat{\mathcal{H}}$. This implies that sampling hyperparameters $\mathcal{H} \setminus \hat{\mathcal{H}}$ can be biased while sampling from $q(\hat{\mathcal{H}})$ is unaffected because the conditions $\boldsymbol{\theta}'_{1,\ldots,N}$ are sampled first in i.i.d. fashion. Our policy selects the configuration evaluated next by uniformly sampling from the remaining $N$ candidates. Since uniformly sampling $L$ times from a set of $N$ samples from a distribution $q$ results in approximating $q$ arbitrarily close for $N \to \infty$ and $L \to \infty$, we conclude that user priors are exactly reflected as specified in our selection policy. This concludes our proof that the selection policy of IBO-HPC is efficacious and feedback-adhering. □

## D.2 IBO-HPC minimizes Simple Regret

We introduce the following proposition:

**Proposition 4** (IBO-HPC minimizes Simple Regret). *IBO-HPC minimizes simple regret, which is defined as $r = f(\boldsymbol{\theta}) - f(\boldsymbol{\theta}^*)$ for a hyperparameter configuration $\boldsymbol{\theta} \in \Theta$ and global optimum $\boldsymbol{\theta}^*$.*

*Proof.* Assume that $w > 0$ holds for each weight $w$ of a PC $s$, that each leaf node of $s$ is a distribution $p$ s.t. $p(x) > 0$ for some $x$ and assume $f$ is not noisy. Then, the PC fulfills the positivity assumption, i.e. $s(\mathcal{H} = \boldsymbol{\theta}, F = f(\boldsymbol{\theta})) > 0$. It follows that $s(\mathcal{H} = \boldsymbol{\theta}|F = f^*) > 0$ for any $f^*$ and any $\boldsymbol{\theta} \in \Theta$. Thus, with iterations $T \to \infty$, the probability of sampling the global optimum $\boldsymbol{\theta}^*$ in one of the iterations gets 1, and thus $r = f(\boldsymbol{\theta}^*) - f(\boldsymbol{\theta}^*) = 0$. □

## D.3 Convergence Behavior of IBO-HPC

In this section, we analyze the convergence behavior of IBO-HPC at each iteration. Therefore, let us state a well-known result of the PC literature on which our analysis is based.

**Definition 4.** *Induced Trees [Zhao et al., 2016]. Given a complete and decomposable PC $s$ over $\mathcal{H} = \{H_1, \ldots, H_n\}$, $\mathcal{T} = (\mathcal{T}_V, \mathcal{T}_E)$ is called an induced tree PC from $s$ if*

1. *$N \in T_V$ where $N$ is the root of $s$.*

2. *for all sum nodes $S \in \mathcal{T}_V$, exactly one child of $S$ in $s$ is in $\mathcal{T}_V$, and the corresponding edge is in $\mathcal{T}_E$.*

3. *for all product node $P \in \mathcal{T}_V$, all children of $P$ in $s$ are in $\mathcal{T}_V$, and the corresponding edges in $\mathcal{T}_E$.*

We can use Def. 4 to represent decomposable and complete PCs as mixtures [Zhao et al., 2016].

**Proposition 5** (Induced Tree Representation). *Let $\tau_s$ be the total number of induced trees in $s$. Then the output at the root of $s$ can be written as $\sum_{t=1}^{\tau_s} \prod_{(k,j) \in \mathcal{T}_{tE}} w_{kj} \prod_{i=1}^{n} p_t(H_i = \boldsymbol{\theta}_i)$, where $\mathcal{T}_t$ is the $t$-th unique induced tree of $s$ and $p_t(H_i)$ is a univariate distribution over $H_i$ in $\mathcal{T}_t$ as a leaf node.*

With this, we are ready to analyze the convergence speed of IBO-HPC in each iteration. Assume a non-noisy differentiable $L$-Lipschitz continuous function $f : \mathbb{R}^d \to \mathbb{R}$ with global optimum $\boldsymbol{\theta}^* \in \mathbb{R}^d$ that is convex within a ball $B_r(\boldsymbol{\theta}^*) = \{\boldsymbol{\theta} \in \mathbb{R}^d : ||\boldsymbol{\theta} - \boldsymbol{\theta}^*|| < r\}$. Further, assume we have given a dataset $\mathcal{D} = \{(\boldsymbol{\theta}_1, y_1), \ldots, (\boldsymbol{\theta}_n, y_n)\}$ where all $\boldsymbol{\theta}_i \in B_r$ and $y_i = f(\boldsymbol{\theta}_i)$ and a decomposable, complete PC $s$ over $\mathcal{H} \cup \{F\}$ where the support of $\mathcal{H} = B_r$ and the support of $F = \mathbb{R}$. Assume $s$ locally maximizes the likelihood over $\mathcal{D}$ and that all leaves are Gaussians. Note that LearnSPN yields decomposable and complete PCs that locally maximize the likelihood of the given data [Gens and Domingos, 2013].

We analyze the convergence properties of our algorithm by examining the expected improvement (EI) in each iteration. Therefore, denote the best score obtained until iteration $t$ as $y_t^*$ and its corresponding configuration as $\boldsymbol{\theta}_t^*$. For better readability, we write $s(\mathcal{H} = \boldsymbol{\theta}|F = y_t^*)$ as $s(\boldsymbol{\theta}|y_t^*)$ from now on. Then, the expected improvement of IBO-HPC within $B_r$ is given by

$$\int_{\boldsymbol{\theta} \in B_r} s(\boldsymbol{\theta}|y_t^*) \cdot \mathbb{I}[f(\boldsymbol{\theta}) < y_t^*] \cdot f(\boldsymbol{\theta}) \tag{5}$$

$$= \int_{\boldsymbol{\theta}^*}^{\boldsymbol{\theta}_t^*} s(\boldsymbol{\theta}|y_t^*) \cdot f(\boldsymbol{\theta}). \tag{6}$$

Here, w.l.o.g. we assume that $\boldsymbol{\theta}_k^* < \boldsymbol{\theta}_{tk}^*$ for all dimensions $k \in \{1, \ldots, d\}$ and call $\mathbb{I}$ the indicator function. Using Prop. 5, the fact that the first product of the induced tree representation of a PC $s$ acts as an edge selector, the fact that the conditional of a PC is a PC again, and the Gaussian leaf parameterization of $s$, we can write $s$ as a Gaussian Mixture, i.e., $s(\boldsymbol{\theta}|y_t^*) = \sum_{i=1}^{\tau_s} w_i \phi(\boldsymbol{\theta}; \boldsymbol{\mu}_i, \Sigma_i)$. Here, $\phi$ is the density of the Gaussian distribution parameterized by mean $\boldsymbol{\mu}$ and covariance matrix $\Sigma$ and corresponds to the second product in the induced tree representation of $s$. Thus, Eq. 5 can be rewritten as

$$\sum_{i=1}^{\tau_s} w_i \int_{\boldsymbol{\theta}^*}^{\boldsymbol{\theta}_t^*} \phi(\boldsymbol{\theta}; \boldsymbol{\mu}_i, \Sigma_i) \cdot f(\boldsymbol{\theta}). \tag{7}$$

Due to the $L$-Lipschitz assumption, $||f(\boldsymbol{\theta}) - f(\boldsymbol{\theta}')|| \leq L \cdot ||\boldsymbol{\theta} - \boldsymbol{\theta}'||$ holds for all $\boldsymbol{\theta}, \boldsymbol{\theta}' \in B_r$. Hence, we can use a Taylor approximation and write $f(\boldsymbol{\theta}) \approx f(\boldsymbol{\theta}^*) + \nabla f(\boldsymbol{\theta}^*) \cdot ||\boldsymbol{\theta} - \boldsymbol{\theta}^*||$ which is upper bounded by $f(\boldsymbol{\theta}^*) + L||\boldsymbol{\theta} - \boldsymbol{\theta}^*||$. Then, we can write an upper bound of EI as

$$\sum_{i=1}^{\tau_s} w_i \int_{\boldsymbol{\theta}^*}^{\boldsymbol{\theta}_t^*} \phi(\boldsymbol{\theta}; \boldsymbol{\mu}_i, \Sigma_i) \cdot (f(\boldsymbol{\theta}^*) + L||\boldsymbol{\theta} - \boldsymbol{\theta}^*||) \tag{8}$$

$$= \sum_{i=1}^{\tau_s} w_i \left( \int_{\boldsymbol{\theta}^*}^{\boldsymbol{\theta}_t^*} \phi(\boldsymbol{\theta}; \boldsymbol{\mu}_i, \Sigma_i) \cdot f(\boldsymbol{\theta}^*) + \int_{\boldsymbol{\theta}^*}^{\boldsymbol{\theta}_t^*} \phi(\boldsymbol{\theta}; \boldsymbol{\mu}_i, \Sigma_i) \cdot L||\boldsymbol{\theta} - \boldsymbol{\theta}^*||\right) \tag{9}$$

$$= \sum_{i=1}^{\tau_s} w_i \cdot \left( f(\boldsymbol{\theta}^*) \cdot \int_{\boldsymbol{\theta}^*}^{\boldsymbol{\theta}_t^*} \phi(\boldsymbol{\theta}; \boldsymbol{\mu}_i, \Sigma_i) + \mathbb{E}_{\phi_i}[g_{\boldsymbol{\theta}^*}(\boldsymbol{\theta})]\right). \tag{10}$$

In the last step, we defined $g_{\boldsymbol{\theta}^*}(\boldsymbol{\theta}) := L||\boldsymbol{\theta} - \boldsymbol{\theta}^*||$. Note that we take the expectation w.r.t. the truncated normal distribution because we consider the interval $[\boldsymbol{\theta}^*, \boldsymbol{\theta}_t^*]$. Also note that $f(\boldsymbol{\theta}^*)$ is constant. Thus, we can omit it for the sake of convergence analysis. Since $g_{\boldsymbol{\theta}^*}$ is linear, we can use the linearity of the expectation and write

$$\sum_{i=1}^{\tau_s} w_i \cdot \left( \int_{\boldsymbol{\theta}^*}^{\boldsymbol{\theta}_t^*} \phi(\boldsymbol{\theta}; \boldsymbol{\mu}_i, \Sigma_i) + g_{\boldsymbol{\theta}^*}(\mathbb{E}_{\phi_i}[\boldsymbol{\theta}])\right) \tag{11}$$

$$= \sum_{i=1}^{\tau_s} w_i \cdot \left( (\Phi(\boldsymbol{\theta}_t^*; \boldsymbol{\mu}_i, \Sigma_i) - \Phi(\boldsymbol{\theta}^*; \boldsymbol{\mu}_i, \Sigma_i)) + L||\mathbb{E}_{\phi_i}[\boldsymbol{\theta}] - \boldsymbol{\theta}^*||\right), \tag{12}$$

where $\Phi(\boldsymbol{\theta}; \boldsymbol{\mu}, \Sigma)$ is the cumulative distribution function of multivariate Gaussian. Since the expectations $\mathbb{E}_{\phi_i}[\boldsymbol{\theta}]$ are taken over the truncated normal, they can be lower bounded by $\boldsymbol{\mu} + \alpha \cdot \text{diag}(\Sigma)$. Thus, we have to set a series of $\alpha_i$ where each $\alpha_i = \min(\boldsymbol{\theta}_t^* - \boldsymbol{\mu}_i, \boldsymbol{\theta}^* - \boldsymbol{\mu}_i)$. Then, we can write

$$\sum_{i=1}^{\tau_s} w_i \cdot \left( (\Phi(\boldsymbol{\theta}_t^*; \boldsymbol{\mu}_i, \Sigma_i) - \Phi(\boldsymbol{\theta}^*; \boldsymbol{\mu}_i, \Sigma_i)) + L||(\boldsymbol{\mu}_i + \alpha_i \cdot \text{diag}(\Sigma_i)) - \boldsymbol{\theta}^*||\right). \tag{13}$$

Setting $\epsilon_i = ||\boldsymbol{\mu}_i + \alpha_i \cdot \text{diag}(\Sigma_i) - \boldsymbol{\theta}^*||$ and splitting the sum yields

$$\sum_{i=1}^{\tau_s} w_i \cdot \Phi(\boldsymbol{\theta}_t^*; \boldsymbol{\mu}_i, \Sigma_i) - \sum_{i=1}^{\tau_s} w_i \cdot \Phi(\boldsymbol{\theta}^*; \boldsymbol{\mu}_i, \Sigma_i) + \sum_{i=1}^{\tau_s} w_i L \epsilon_i. \tag{14}$$

Using that the cumulative multivariate Gaussian $\Phi(\boldsymbol{\theta}_t^*; \boldsymbol{\mu}_i, \Sigma_i)$ can be lower bounded by $\prod_{j=1}^{d} \Phi(\boldsymbol{\theta}_{ti}^*; \boldsymbol{\mu}_{ij}, \Sigma_{i_{jj}})$, we can lower-bound the entire equation, giving us

$$\sum_{i=1}^{\tau_s} w_i \cdot \prod_{j=1}^{d} \Phi(\boldsymbol{\theta}_{tj}^*; \boldsymbol{\mu}_{ij}, \Sigma_{i_{jj}}) - \sum_{i=1}^{\tau_s} w_i \cdot \prod_{j=1}^{d} \Phi(\boldsymbol{\theta}_j^*; \boldsymbol{\mu}_{ij}, \Sigma_{i_{jj}}) + \sum_{i=1}^{\tau_s} w_i L \epsilon_i. \tag{15}$$

Since $\Phi(\frac{x-\mu}{\sigma}) = \frac{1}{2}\left(1 + \text{erf}(\frac{x-\mu}{\sigma\sqrt{2}})\right)$ holds, we rewrite

$$\sum_{i=1}^{\tau_s} w_i \cdot \prod_{j=1}^{d} \text{erf}\left(\frac{\boldsymbol{\theta}_{tj}^* - \boldsymbol{\mu}_{ij}}{\Sigma_{i_{jj}}\sqrt{2}}\right) - \sum_{i=1}^{\tau_s} w_i \cdot \prod_{j=1}^{d} \text{erf}\left(\frac{\boldsymbol{\theta}_j^* - \boldsymbol{\mu}_{ij}}{\Sigma_{i_{jj}}\sqrt{2}}\right) + \sum_{i=1}^{\tau_s} w_i L \epsilon_i \tag{16}$$

$$= \sum_{i=1}^{\tau_s} w_i \cdot \left(\prod_{j=1}^{d} \text{erf}\left(\frac{\boldsymbol{\theta}_{tj}^* - \boldsymbol{\mu}_{ij}}{\Sigma_{i_{jj}}\sqrt{2}}\right) - \prod_{j=1}^{d} \text{erf}\left(\frac{\boldsymbol{\theta}_j^* - \boldsymbol{\mu}_{ij}}{\Sigma_{i_{jj}}\sqrt{2}}\right) + L \epsilon_i\right). \tag{17}$$

Note that we dropped constants and scaling by $\frac{1}{2}$ of the error function as it does not affect the overall result.

Intuitively spoken, the EI is lower bounded by the cumulative probability mass (given by error function erf) within the region defined by the largest discrepancy between minimal error w.r.t. to the observed data (i.e., bad convergence when $s$ overfits) and the maximal error w.r.t. $\boldsymbol{\theta}^*$ (i.e., $\mathcal{D}$ does not contain points close to the optimum), multiplied by a linear approximation of the objective $f$ between the best observed configuration $\boldsymbol{\theta}_t^*$ and $\boldsymbol{\theta}^*$.

Note that this result does not incorporate user knowledge. The analysis of the effect of user knowledge is straightforward. If helpful user knowledge is given, this can be seen as shifting at least one dimension $j$ of at least one mean vector $\boldsymbol{\mu}_k$ by some $\delta$ towards $\boldsymbol{\theta}^*$, i.e., $\boldsymbol{\mu}_{*k} = \boldsymbol{\mu}_k + (0, \ldots, \delta, \ldots, 0)$. Then, assuming all $\Sigma_i$ stay as above,

$$\sum_{i=1}^{\tau_s} w_i \cdot \left(\prod_{j=1}^{d} \text{erf}\left(\frac{\boldsymbol{\theta}_{tj}^* - \boldsymbol{\mu}_{ij}}{\Sigma_{i_{jj}}\sqrt{2}}\right) - \prod_{j=1}^{d} \text{erf}\left(\frac{\boldsymbol{\theta}_j^* - \boldsymbol{\mu}_{ij}}{\Sigma_{i_{jj}}\sqrt{2}}\right) + L \epsilon_i\right)$$

$$\leq \sum_{i=1, i \neq k}^{\tau_s} w_i \cdot \left(\prod_{j=1}^{d} \text{erf}\left(\frac{\boldsymbol{\theta}_{tj}^* - \boldsymbol{\mu}_{ij}}{\Sigma_{i_{jj}}\sqrt{2}}\right) - \prod_{j=1}^{d} \text{erf}\left(\frac{\boldsymbol{\theta}_j^* - \boldsymbol{\mu}_{ij}}{\Sigma_{i_{jj}}\sqrt{2}}\right) + L \epsilon_i\right)$$

$$+ w_k \cdot \left(\prod_{j=1}^{d} \text{erf}\left(\frac{\boldsymbol{\theta}_{tj}^* - \boldsymbol{\mu}_{kj}}{\Sigma_{k_{jj}}\sqrt{2}}\right) - \prod_{j=1}^{d} \text{erf}\left(\frac{\boldsymbol{\theta}_j^* - \boldsymbol{\mu}_{kj}}{\Sigma_{k_{jj}}\sqrt{2}}\right) + L \epsilon_k\right).$$

This is easy to see since the distribution we sample configurations from is shifted towards the global optimum $\boldsymbol{\theta}^*$, thus increasing the probability of sampling a configuration closer to $\boldsymbol{\theta}^*$, ultimately leading to faster convergence.

## D.4 Accuracy of IBO-HPC's Selection Policy

Here, we briefly discuss the accuracy of IBO-HPC's policy in selecting new configurations for evaluation based on the obtained data (see Eq. 2). Note that the sampling from the distribution provided in Eq. 2 is accurate if (1) the $s$ represents the data $\mathcal{D}$ accurately and (2) sampling from $s$ and

the prior $q$ is unbiased (i.e., samples are drawn according to the underlying distribution). Let us start with (1). Since we employ LearnSPN [Gens and Domingos, 2013] to obtain $s$ (a PC in form of SPN), $s$ will locally maximize the log-likelihood of the training data (i.e., the configuration-evaluation pairs obtained). This means that there is no other SPN in the space of the learnable SPNs via LearnSPN that achieves a better log-likelihood given the data.[1] Hence, as long as the ground truth distribution $p$ (or a good approximation of it) is representable by an SPN, we can recover $p$ with arbitrarily small error with iterations $T \to \infty$.

Looking at (2), we sample from two distributions when selecting a new configuration. First, we sample from the prior $q$, then from the conditional $s(\mathcal{H} \setminus \hat{\mathcal{H}} | \hat{\mathcal{H}} = \boldsymbol{\theta}, F = f^*)$ where $\boldsymbol{\theta} \sim q$. Assuming $q$ is a tractable distribution (e.g., a parametric one such as an isotropic Gaussian), sampling is immediate and not biased (i.e., performed via simple transformations such as the Box-Muller transform). Note that the assumption on $q$ being a tractable (and relatively simple) distribution can be made safely since providing highly complex distributions as user knowledge is hard to do for most users. When considering sampling from the conditional $s(\mathcal{H} \setminus \hat{\mathcal{H}} | \hat{\mathcal{H}} = \boldsymbol{\theta}, F = f^*)$, it should be noted that this conditional is a valid PC again (specifically, a PC in the form of an SPN when obtained with LearnSPN). The model is unchanged and only evaluated differently, i.e., by providing the partial evidence at leaves and evaluating the model bottom-up first (see Choi et al. [2020]). Then, PC sampling is performed top-down by sampling from the simple categorical variables represented by the sum nodes and then from the selected univariate leaves. Thus, the process is tractable (linear in the circuit size) and not biased by further operations or assumptions [Choi et al., 2020]. Thus, we conclude that the approximation in Eq. 2 is accurate in the limit $N, T \to \infty$.

## E  Experimental Details

Here, we present additional details of our empirical evaluation. The raw logs from our experiments are available at `https://figshare.com/ndownloader/files/53323751`, and our code is available at `https://github.com/ml-research/ibo-hpc`.

### E.1  Benchmarks

Benchmarks are a valuable tool in HPO/NAS research. They allow a cheap and reproducible evaluation of HPO and NAS algorithms, thus allowing researchers without many computing resources to test their algorithms reliably under real-world scenarios. Benchmarks achieve cheap evaluation and reproducibility by providing pre-computed training and evaluation statistics for a set of tasks. A task is usually defined by a dataset (e.g., CIFAR-10) and a search space (e.g., a space over neural architectures). Then, all configurations of the search space (or a large part of it) are evaluated on the given dataset(s) and the results are saved in a look-up table. Some benchmarks go beyond mere look-up tables and train a surrogate model on the saved results to predict e.g., the accuracy of a model given a configuration. This is especially useful for continuous hyperparameters since a look-up table only represents a discrete subset of the space while the surrogate model can interpolate between values.

In our experimental evaluation, we used the following benchmarks: JAHS [Bansal et al., 2022], NAS-Bench-101 [Ying et al., 2019], NAS-Bench-201 [Dong and Yang, 2020], HPO-B [Pineda-Arango et al., 2021], FCNet [Klein and Hutter, 2019], and PD1 [Wang et al., 2024]. We now briefly describe the characteristics of these benchmarks.

**JAHS**. JAHS aims to provide a reproducible benchmark for joint optimization of neural architectures and other hyperparameters on real-world tasks. It consists of a 14-dimensional search space (of which some dimensions, like the number of epochs, can also be treated as fidelities). The search space contains both discrete and continuous hyperparameters. Regarding tasks, JAHS provides

---

[1]Assuming an oracle for the variable splitting. See Proposition 1 in Gens and Domingos [2013].

training and evaluation statistics of models trained on CIFAR-10, Fashion-MNIST and Colorectal Histology. All tasks are image classification tasks. See [Bansal et al., 2022] for more information.

**NAS-Bench-101/201**. NAS-Bench-101/201 provide a reproducible benchmark for neural architecture search. Both come with similar search spaces. However, they use different encoding to represent architectures. Thus, NAS-Bench-101 comes with a sparser but high-dimensional search space (26 dimensions), while NAS-Bench-201 comes with a more dense encoding and a 6-dimensional search space. NAS-Bench-101 provides training and evaluation statistics for CIFAR-10, while NAS-Bench-201 provides those statistics for CIFAR-10, CIFAR-100, and Imagenet. In our experiments, we only used CIFAR-10. See [Ying et al., 2019, Dong and Yang, 2020] for more information.

**HPO-B**. HPO-B is a large-scale HPO benchmark based on a diverse set of OpenML tasks. It comes with 176 search spaces and 196 datasets. Since some search spaces have been evaluated on multiple datasets, many tasks are available. Most of the tasks are classification and regression tasks on different data modalities, such as tabular data or image data. In our evaluation, we used the credit-g and vehicle datasets (dataset IDs 31 and 9914), paired with search spaces over hyperparameters of a random search model (search space ID 6794) and a gradient boosting model (search space ID 6767). Both datasets are tabular datasets. We chose these since both provide many evaluations reported at OpenML (506k for credit-g and 31k for vehicle), allowing rigorous comparability. The search spaces we used contained discrete and continuous hyperparameters. The gradient boosting search space is defined over 17 hyperparameters, while the random forest search space is defined over 9 hyperparameters. For more information, refer to [Pineda-Arango et al., 2021].

**PD1**. The PD1 benchmark is an HPO benchmark for neural networks. The search space is defined over 4 continuous hyperparameters (learning rate, momentum, learning rate decay, and learning rate decay steps) that influence the learning behavior of the networks. Since the search space does not contain any architecture-specific hyperparameters, PD1 is not a NAS benchmark, although it only considers neural networks as models. Different variants of CNNs and Transformer architectures have been evaluated on eight different tasks spanning image classification and language modeling and corresponding training and evaluation statistics are provided. In our experiments, we considered the HPO task defined by PD1 with a ResNet50 as a neural network and Imagenet as a large-scale image classification task. In contrast to CIFAR-10, Imagenet is more realistic as it is more diverse than CIFAR-10. For more information, refer to [Wang et al., 2024].

**FCNet**. FCNet provides training and evaluation statistics for fully connected neural networks, evaluated on four different tabular regression tasks. The search space contains 9 dimensions, including some architecture choices (activation function, number of hidden units) and optimization-specific hyperparameters such as the learning rate. While architecture choices are naturally discrete, FCNet has discretized all continuous hyperparameters. Thus, all hyperparameters are discrete in the FCNet search space. In our experiments, we chose the SLICE_LOCALIZATION task as it is the most challenging task included in FCNet regarding feature dimension (385) and number of available samples (31k). Refer to [Klein and Hutter, 2019] for more details.

### E.2 Search Space Extension of JAHS

To make the HPO problem on JAHS more challenging, we decided to extend the search space slightly as JAHS – as a surrogate benchmark – allows us to query hyperparameter values which were not tested explicitly in the benchmark. We defined three search spaces for JAHS which are presented in the following table.

|                  | S1                         | S2                         | S3                         |
|------------------|----------------------------|----------------------------|----------------------------|
| Activation       | [Mish, ReLU, Hardswish]    | [Mish, ReLU, Hardswish]    | [Mish, ReLU, Hardswish]    |
| Learning Rate    | [1e-3, 1e0]                | [1e-3, 1e0]                | [1e-3, 1e0]                |
| Weight Decay     | [1e-5, 1e-2]               | [1e-5, 1e-2]               | [1e-5, 1e-2]               |
| Trivial Argument | [True, False]              | [True, False]              | [True, False]              |
| Op1              | 0-6                        | 0-6                        | 0-6                        |
| Op2              | 0-6                        | 0-6                        | 0-6                        |
| Op3              | 0-6                        | 0-6                        | 0-6                        |
| Op4              | 0-6                        | 0-6                        | 0-6                        |
| Op5              | 0-6                        | 0-6                        | 0-6                        |
| Op6              | 0-6                        | 0-6                        | 0-6                        |
| N                | 1-15                       | 1-11                       | 1-5                        |
| W                | 1-31                       | 1-23                       | 1-16                       |
| Epoch            | 1-200                      | 1-200                      | 1-200                      |
| Resolution       | 0-1                        | 0-1                        | 0-1                        |

Table 1: **JAHS Search Space**. We define three versions of the JAHS search space, ranging from simpler to harder spaces.

### E.3 Interactions

Here, we provide the interactions used in our experiments. For the experiments, beneficial and misleading user interactions have been defined as user priors for each benchmark. We aim to analyze the behavior of IBO-HPC under various user interactions. Thus, we analyze several scenarios in which users interact with IBO-HPC: (1) The user provides beneficial knowledge about only a few hyperparameters in early optimization iterations. (2) The user provides beneficial knowledge about only a few hyperparameters at later iterations. By providing beneficial knowledge only for a small set of hyperparameters, we analyze whether IBO-HPC can effectively leverage knowledge that helps it converge, even if this kind of knowledge is only sparse. (3) The user provides misleading knowledge for many hyperparameters, thus challenging our recovery mechanism in cases in which IBO-HPC is misled over the majority of dimensions. (4) The user interacts multiple times, alternating between providing beneficial and misleading knowledge. This tests if IBO-HPC can handle contradictory sets of knowledge given during optimization, thus analyzing if IBO-HPC handles interactions that could occur in the real world well. With the alternation between beneficial and misleading knowledge, we can analyze whether IBO-HPC effectively makes use of beneficial knowledge while the recovery mechanism prevents IBO-HPC from stalling when misleading user knowledge is provided.

To define beneficial and misleading user knowledge/priors for each benchmark and the corresponding search space, we randomly sample $10k$ configurations and kept the best/worst performing ones, denoted as $\theta^+$ and $\theta^-$, respectively. To demonstrate that beneficial user priors over a few hyperparameters are enough to improve the performance of IBO-HPC remarkably, we define beneficial interactions by selecting a small subset of hyperparameters $\hat{\mathcal{H}} \subset \mathcal{H}$. The subset size was set to cover less than 25% of the hyperparameters. The subset was sampled randomly and was reused for all experiments. Then, we define a prior over each $H \in \hat{\mathcal{H}}$ favoring the value of $H$ given in $\theta^+$, denoted by $\theta^+[H]$, with a probability up to 1000 times higher than for other values. For misleading interaction, $\hat{\mathcal{H}}$ is chosen to be large to demonstrate that IBO-HPC recovers even if a large amount of misleading information is provided. This time, the probability to sample $\theta^-[H]$ is up to 1000 times higher than for other values for each $H \in \hat{\mathcal{H}}$. The priors are chosen to be rather strong since, as emphasized in Sec. 1, the stronger the prior, the better $\pi$BO and BOPrO reflect user knowledge in their selection policy. Striving for a fair comparison, we opt for such strong priors. Further, we aim to show that IBO-HPC reliably recovers from receiving large amounts of strongly misleading

knowledge. Sometimes, users might want to specify a concrete value for certain hyperparameters instead of defining a distribution. Thus, we also conducted experiments with point masses as user priors. For the experiments in which multiple interactions are provided, we randomly chose a beneficial and a misleading interaction and provided them to IBO-HPC alternatingly.

**JAHS**. The following JSON code shows the interactions performed in our JAHS experiments. The first interaction is a misleading interaction, followed by a beneficial interaction and a no interaction (for recovery).

```
[
    {
        "type": "bad",
      "intervention": {"Activation": 1, "LearningRate": 0.8201676371308472, "N": 15,
        "Op1": 3, "Op2": 4, "Op3": 1, "Op4": 2, "Resolution": 0.5096959403985494,
        "TrivialAugment": 0, "W": 14,
         "WeightDecay": 0.002697686639935806, "epoch": 10},
        "iteration": 5
    },
    {
        "type": "good",
        "intervention": {"N": 3, "W": 16, "Resolution": 1},
        "iteration": 15
    },
    {
        "type": "good",
        "intervention": null,
        "iteration": 20
    },
    {
        "type": "good",
        "kind": "dist",
        "intervention": {"N": {"dist": "cat", "parameters":
        [1, 1, 1, 1e4, 1, 1, 1, 1, 1, 1, 1, 1, 1, 1, 1, 1]},
        "W": {"dist": "cat", "parameters":
        [1, 1, 1, 1, 1, 1, 1, 1, 1, 1, 1, 1, 1, 1, 1, 1, 1e4]},
        "Resolution": {"dist": "uniform", "parameters": [0.98, 1.02]}},
        "iteration": 5
    }
]
```

**NAS-Bench-101**. The following JSON code shows the interactions performed in our experiments on NAS-Bench-101. The first interaction is a misleading interaction, followed by a beneficial interaction and a no interaction (for recovery).

```
[
    {
        "type": "bad",
        "kind": "point",
      "intervention": [0, 1, 1, 0, 0, 0, 0, 1, 0, 0, 0, 1, 0, 1, 0, 0, 1, 1, 1, 0, 1],
        "iteration": 5
    },
    {
```

```
        "type": "good",
        "kind": "point",
     "intervention": [1, 0, 1, 0, 1, 1, 1, 0, 0, 0, 0, 0, 1, 0, 0, 0, 1, 0, 1, 0, 1],
        "iteration": 12
    },
    {
        "type": "good",
        "kind": "point",
        "intervention": null,
        "iteration": 20
    },
    {
        "type": "good",
        "kind": "dist",
        "intervention": {
            "e_0_1": {"dist": "cat", "parameters": [1, 1e4]},
            "e_0_2": {"dist": "cat", "parameters": [1e4, 1]},
            "e_0_3": {"dist": "cat", "parameters": [1, 1e4]},
            "e_0_4": {"dist": "cat", "parameters": [1e4, 1]},
            "e_0_5": {"dist": "cat", "parameters": [1, 1e4]},
            "e_0_6": {"dist": "cat", "parameters": [1, 1e4]},
            "e_1_2": {"dist": "cat", "parameters": [1, 1e4]},
            "e_1_3": {"dist": "cat", "parameters": [1e4, 1]},
            "e_1_4": {"dist": "cat", "parameters": [1e4, 1]},
            "e_1_5": {"dist": "cat", "parameters": [1e4, 1]},
            "e_1_6": {"dist": "cat", "parameters": [1e4, 1]},
            "e_2_3": {"dist": "cat", "parameters": [1e4, 1]},
            "e_2_4": {"dist": "cat", "parameters": [1, 1e4]},
            "e_2_5": {"dist": "cat", "parameters": [1e4, 1]},
            "e_2_6": {"dist": "cat", "parameters": [1e4, 1]},
            "e_3_4": {"dist": "cat", "parameters": [1e4, 1]},
            "e_3_5": {"dist": "cat", "parameters": [1, 1e4]},
            "e_3_6": {"dist": "cat", "parameters": [1e4, 1]},
            "e_4_5": {"dist": "cat", "parameters": [1, 1e4]},
            "e_4_6": {"dist": "cat", "parameters": [1e4, 1]},
            "e_5_6": {"dist": "cat", "parameters": [1, 1e4]}
        },
        "iteration": 5
    }
]
```

**NAS-Bench-201.** The following JSON code shows the interactions performed in our experiments on NAS-Bench-201. The first interaction is a misleading interaction, followed by a beneficial interaction and a no interaction (for recovery).

```
[
    {
        "type": "good",
        "kind": "point",
        "intervention": {"Op_0": 2, "Op_1": 2, "Op_2": 0},
```

```
        "iteration": 5
    },
    {

        "type": "bad",
        "kind": "point",
        "intervention": {"Op_0": 1, "Op_1": 2, "Op_2": 1},
        "iteration": 5
    },
    {

        "type": "good",
        "kind": "point",
        "intervention": null,
        "iteration": 20
    },
    {
        "type": "good",
        "kind": "dist",
        "intervention": {"Op_0": {"dist": "cat", "parameters": [1, 1, 1e4, 1, 1]},
                         "Op_1": {"dist": "cat", "parameters": [1, 1, 1e4, 1, 1]},
                         "Op_2": {"dist": "cat", "parameters": [1e4, 1, 1, 1, 1]}},
        "iteration": 5
    }
]
```

**HPO-B**. The following shows the interactions defined for our HPO-B experiments. Each of the three interactions corresponds to one of the three tasks we tested from HPO-B. The order is 6767:31, 6794:31, 6794:9914. Note that we only applied beneficial interactions in the case of HPO-B.

```
    {
        "type": "good",
        "kind": "dist",
        "intervention": {
            "eta": {"dist": "uniform", "parameters": [0.45, 0.55]},
            "subsample": {"dist": "uniform", "parameters": [0.55, 0.65]},
            "lambda": {"dist": "uniform", "parameters": [-150, 0]},
            "min_child_weight": {"dist": "uniform", "parameters": [-15, -10]}},
        "iteration": 1
    },
    {
        "type": "good",
        "kind": "dist",
        "intervention": {
            "num_trees": {"dist": "int_uniform", "parameters": [1690, 1720]},
            "mtry": {"dist": "int_uniform", "parameters": [30, 34]},
            "min_node_size": {"dist": "int_uniform", "parameters": [780, 800]}
        },
        "iteration": 1
    },
    {
```

```
        "type": "good",
        "kind": "dist",
        "intervention": {
            "num_trees": {"dist": "int_uniform", "parameters": [1690, 1720]},
            "mtry": {"dist": "int_uniform", "parameters": [280, 320]},
            "sample_fraction": {"dist": "uniform", "parameters": [0.5, 0.53]}
        },
        "iteration": 1
    }
```

**PD1**. The following shows the interactions defined for the task considered on PD1. Note that we only applied beneficial interactions in the case of PD1.

```
    {
        "type": "good",
        "kind": "dist",
        "intervention": {
            "hps.lr_hparams.decay_steps_factor": {"dist": "uniform",
                                                  "parameters": [0.8, 1.0]},
            "hps.opt_hparams.momentum": {"dist": "uniform",
                                                  "parameters": [1.0, 1.2]}
        },
        "iteration": 1
    }
```

**FCNet**. The following shows the interactions defined for the task considered on FCNet. Note that we only applied beneficial interactions in the case of FCNet.

```
    {
        "type": "good",
        "kind": "dist",
        "intervention": {
            "activation_fn_1": {"dist": "cat", "parameters": [100, 1],
                                "values": ["relu", "tanh"]},
            "activation_fn_2": {"dist": "cat", "parameters": [1, 100],
                                "values": ["relu", "tanh"]},
            "n_units_1": {"dist": "cat", "parameters": [1, 1, 1, 1, 1, 100],
                                "values": [16, 32, 64, 128, 256, 512]},
            "n_units_2": {"dist": "cat", "parameters": [1, 1, 1, 1, 1, 100],
                                "values": [16, 32, 64, 128, 256, 512]}
        },
        "iteration": 1
    }
```

### E.4 Further Results & Ablations

In this section, we provide further results and ablations. Fig. 6 provides additional results on five challenging tasks of the HPO-B, PD1 and FCNet benchmarks. For HPO-B, we used a random forest HPO problem (search space ID 6767) and a gradient-boosting HPO problem (search space ID 6794). As datasets, we used credit-g (dataset ID 31) and vehicle (9914) since both are widely used benchmark datasets for classification. Regarding PD1, we used a ResNet50 as an architecture and optimized

over the 4-dimensional search space provided by PD1. As a dataset, we considered Imagenet, a challenging real-world classification task. Finally, for FCNet, we used the slice localization dataset since it is the most complex dataset in the FCNet benchmark w.r.t. the number of features (385) and number of data instances. IBO-HPC outperforms the baselines or is competitive with the baselines in both cases, i.e., where feedback is given, and no user feedback is given.

Fig 7 shows results of IBO-HPC on JAHS, NAS201, and NAS101 where the given user feedback was either a fixed value or a distribution over configurations. Both cases are handled well by IBO-HPC, demonstrating its flexibility. Fig. 8 provides a more detailed view of the effectiveness of IBO-HPC and its recovery mechanism. It can be seen that IBO-HPC successfully recovers from harmful user feedback in JAHS and NAS-201 (—). Also, it can be seen that IBO-HPC handles alternating and contradictory user feedback well by leveraging information from beneficial feedback and ignoring harmful feedback (—). In NAS-101, however, IBO-HPC is less effective in general, which can be explained by the extreme sparsity of the NAS-101 benchmark. While NAS-101 and NAS-201 are highly similar, NAS-101 uses a binary encoding of architectures, while NAS-201 uses a much denser dictionary-like representation. Although both benchmarks are highly similar, IBO-HPC performs well on NAS-201 but is not as effective on NAS-101, underlining our explanation.

Fig. 9 and 10, we show the CDF of test accuracy/mean squared error (MSE) across the baselines and IBO-HPC. It can be seen that IBO-HPC invests most of the computational resources in good-performing configurations. In other words, IBO-HPC avoids exploration in unpromising regions of the search space. This is because IBO-HPC samples configurations from a conditional distribution where the condition is the best evaluation score obtained. Thus, exploration is purely data-driven and focuses on regions that perform similarly to the incumbent at a particular iteration.

Fig. 11 shows the influence of the decay parameter $\gamma$ in cases where harmful or misleading user knowledge was provided to IBO-HPC at an early iteration (10 in this case). It can be seen that for higher $\gamma$, IBO-HPC requires more time to recover than for smaller $\gamma$. This aligns with our expectations since a larger $\gamma$ corresponds to a high likelihood of the user knowledge being used for many iterations. In contrast, if $\gamma$ is small, likely, the user knowledge is only considered for a certain number of iterations with high likelihood. Thus, for smaller $\gamma$ IBO-HPC can recover faster.

Fig. 12 shows the effect of conditioning on the $\{0.25, 0.5, 0.75\}$-quantile of the obtained evaluation scores instead of the maximum evaluation score. As expected, the higher the quantile, the better the performance of IBO-HPC as we aim to maximize the objective function. Thus, conditioning on higher values guides the optimization algorithm to configurations that yield better evaluation scores.

Lastly, Fig. 13 depicts the effect of changing $L$, i.e. the number of samples drawn from the surrogate before the surrogate is updated. We found that the sample size has no effect on the overall performance of IBO-HPC. However, for some tasks (JAHS CIFAR-10 and CO), a significant variation of convergence speed in early iterations – depending on the choice of $L$ – was obtained. Choosing $L = 5$ seems to lead to fast and stable convergence behavior. Note that setting $L = 1$ would mean that we re-fit the PC in each iteration. However, this also linearly increases the cost of optimization, which is not desirable in practice.

We followed the same experimental protocol as for all other experiments in Fig. 11-13, except that each algorithm was run only 100 times instead of 500 times on each task.

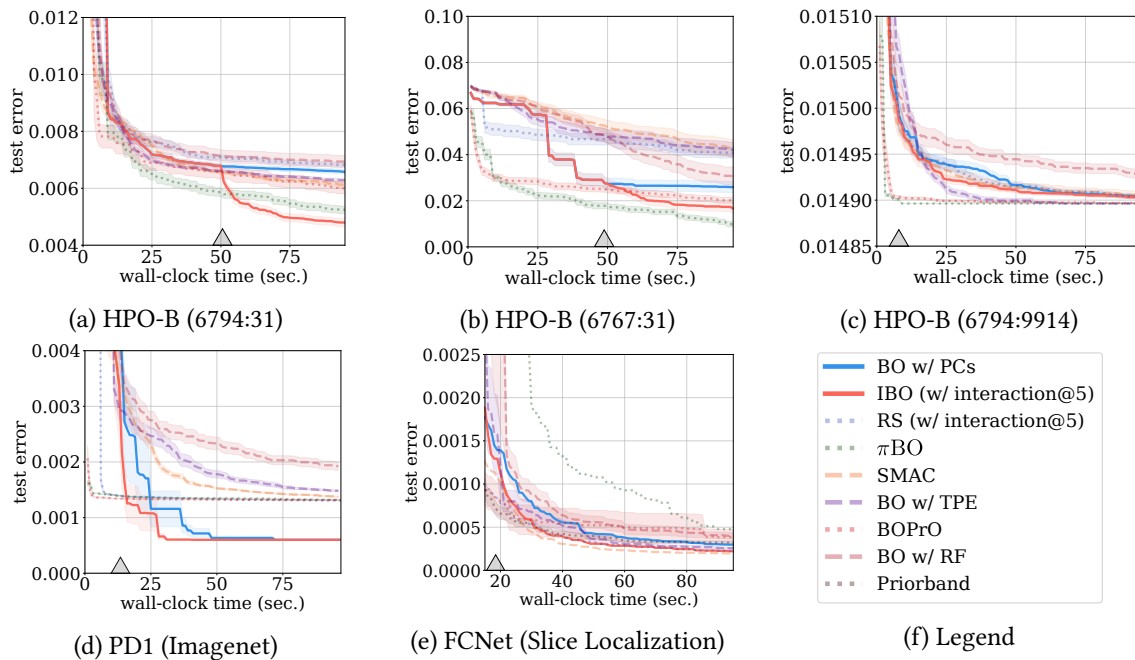

(a) HPO-B (6794:31)  (b) HPO-B (6767:31)  (c) HPO-B (6794:9914)

(d) PD1 (Imagenet)  (e) FCNet (Slice Localization)  (f) Legend

Figure 6: **IBO-HPC is competitive or outperforms strong baselines on HPO-B, PD1 and FCNet**. IBO-HPC outperforms all BO baselines that allow users to provide a prior before optimization when feedback is provided at the 2nd iteration on 3/5 tasks. For the other tasks, only one baseline ($\pi$BO in (b)) beats IBO-HPC or IBO-HPC is on par with the baselines (c). Moreover, IBO-HPC is competitive with other BO methods without any user knowledge given. Results were obtained on HPO-B search spaces 6794 and 6767 with dataset IDs 31 and 9914. For PD1, we optimized hyperparameters of a ResNet50 training on Imagenet, and for FCNet, we tuned hyperparameters of a fully connected neural network on SLICE_LOCALIZATION.

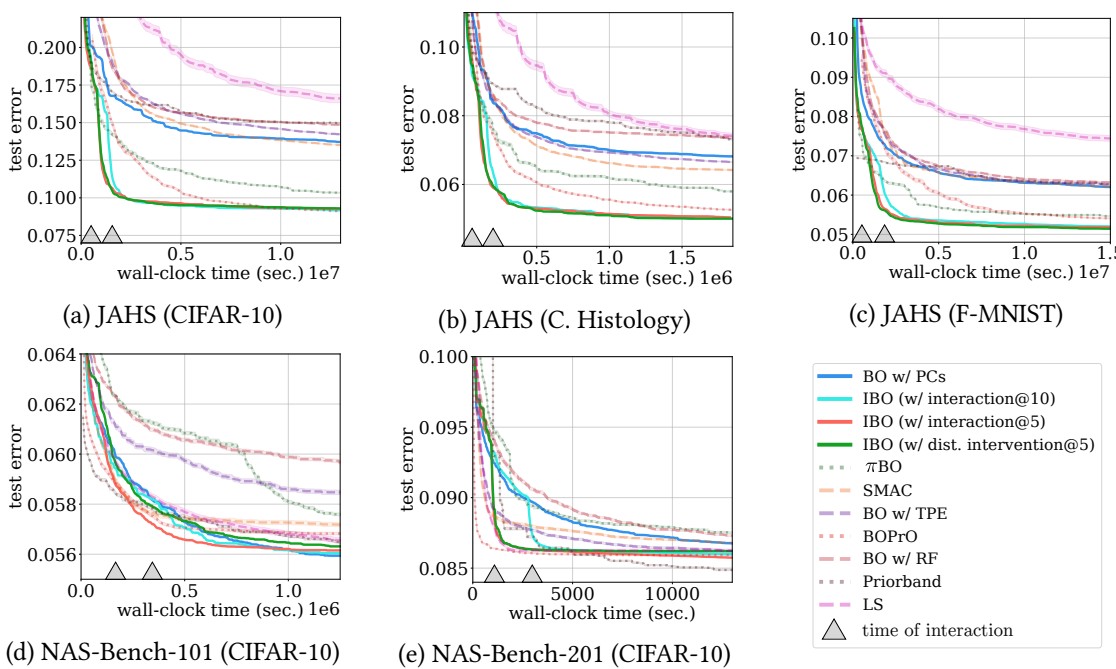

Figure 7: **IBO-HPC outperforms state of the art**. For 4/5 tasks across three challenging benchmarks, IBO-HPC is competitive with strong baselines when no user knowledge is provided. When beneficial user beliefs (△) are provided, either as distributions (——) or point values (——, ——), it outperforms all competitors w.r.t. convergence and solution quality on most tasks. Early interactions (——/—— at 5th iteration, —— at 10th iteration) speed convergence up.

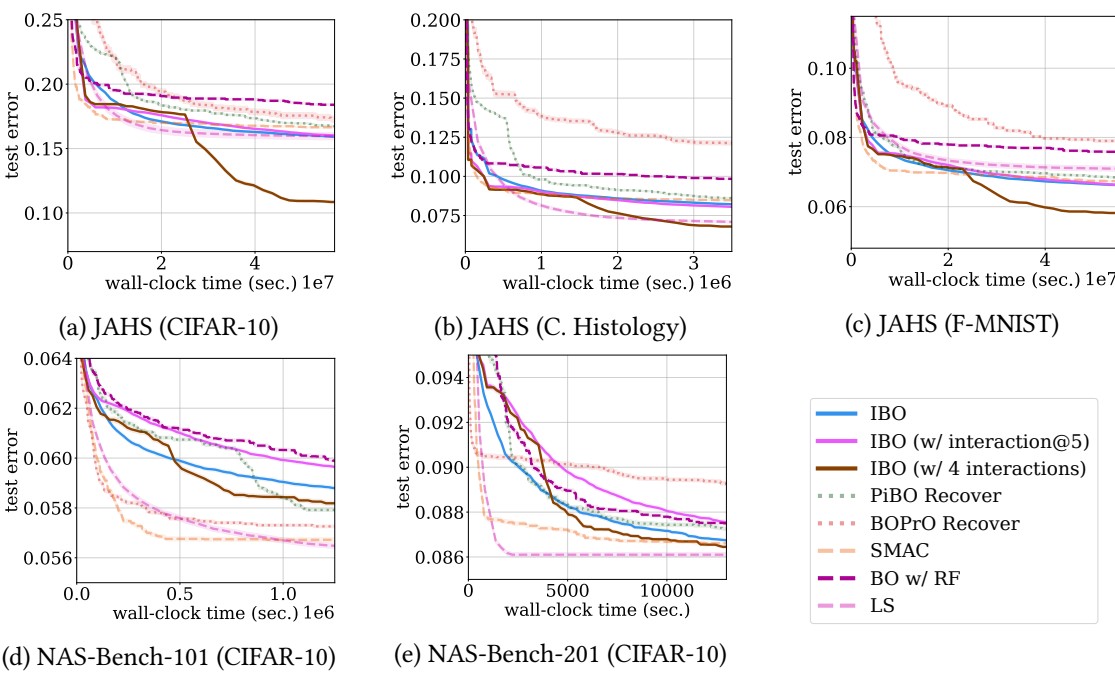

Figure 8: **IBO-HPC recovers from misleading user feedback**. IBO-HPC successfully and consistently recovers from misleading user feedback and performs equally well as if no feedback was given. Also, IBO-HPC handles alternating, contradictory feedback well and is able to leverage beneficial feedback while ignoring misleading feedback.

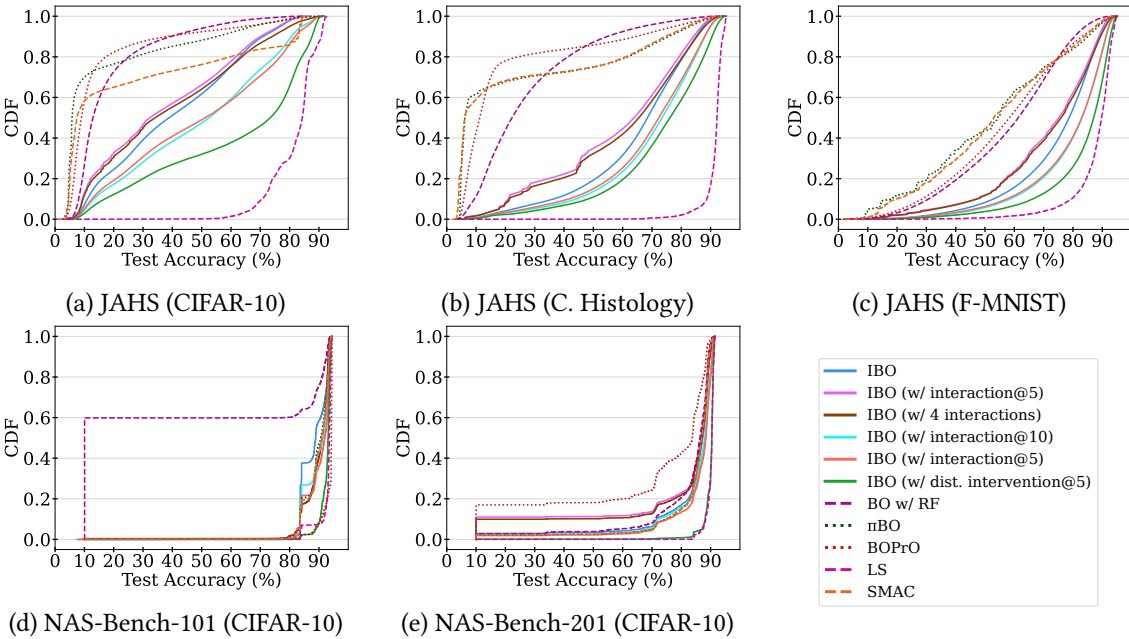

Figure 9: **CDF of Test Accuracy**. The majority of IBO-HPC's sampled candidate configurations are high-performing configurations. Thus, IBO-HPC invests more computational resources in good configurations than other methods. We conjecture that this is because IBO-HPC selects configurations s.t. they are likely to perform similarly to the incumbent in each iteration. Interestingly, RS also samples many well-performing configurations on the JAHS benchmark.

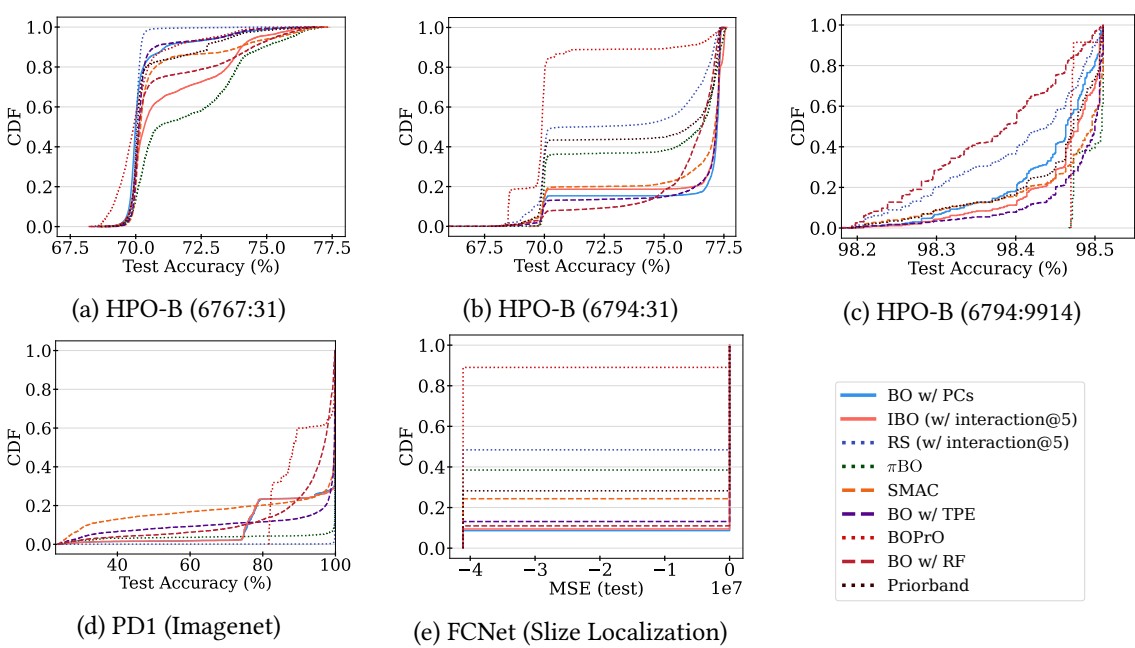

Figure 10: **CDF of Test Accuracy/Mean Squared Error (MSE)**. The majority of IBO-HPC's sampled candidate configurations are high-performing configurations. Thus, IBO-HPC invests more computational resources in good configurations than other methods. We conjecture that this is because IBO-HPC selects configurations s.t. they are likely to perform similarly to the incumbent in each iteration.

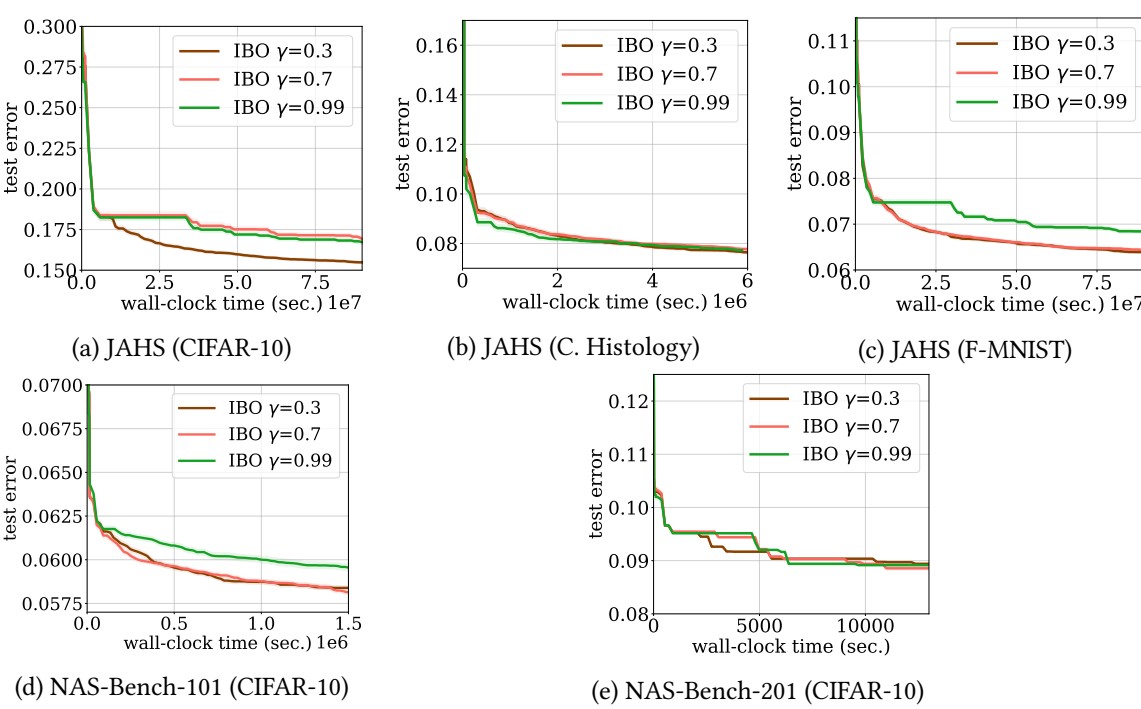

(a) JAHS (CIFAR-10)  (b) JAHS (C. Histology)  (c) JAHS (F-MNIST)

(d) NAS-Bench-101 (CIFAR-10)  (e) NAS-Bench-201 (CIFAR-10)

Figure 11: **Ablation: Effect of $\gamma$ on recovery of IBO-HPC**. As expected, we found that IBO-HPC recovers faster for smaller values of $\gamma$. This is because smaller $\gamma$ values lead to a higher decay of the probability of conditioning on the provided user knowledge. Thus, with faster decay, IBO-HPC recovers faster from harmful or misleading user knowledge (provided at iteration 10).

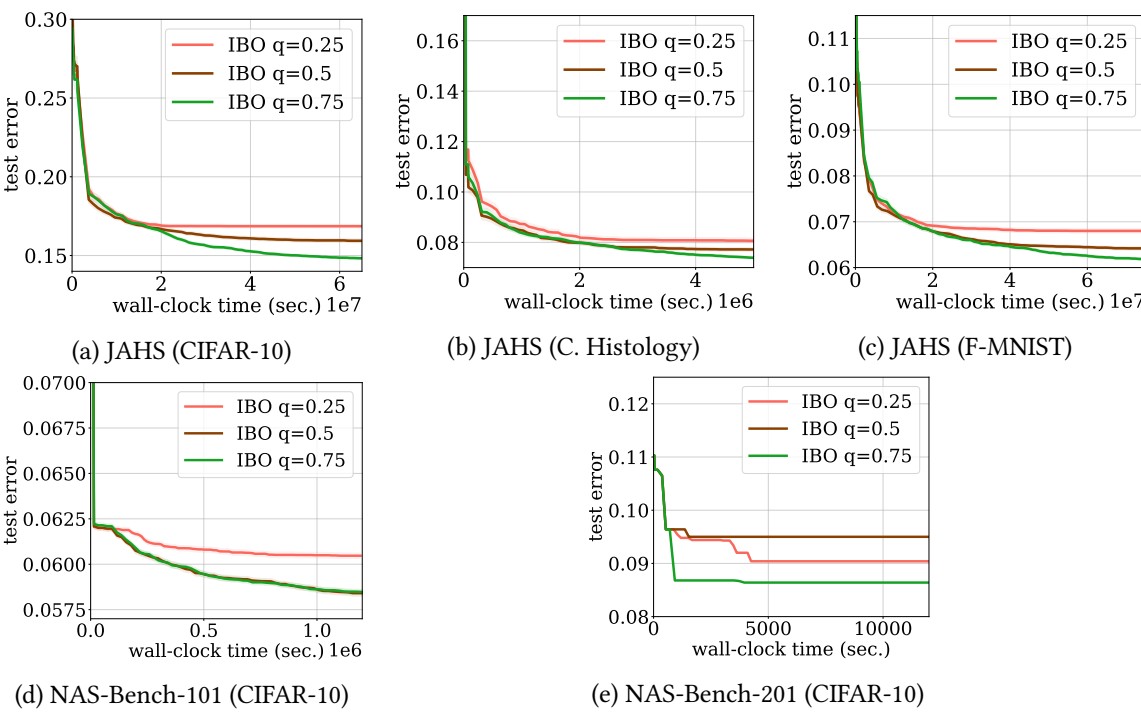

(a) JAHS (CIFAR-10)  (b) JAHS (C. Histology)  (c) JAHS (F-MNIST)

(d) NAS-Bench-101 (CIFAR-10)  (e) NAS-Bench-201 (CIFAR-10)

Figure 12: **Conditioning on sub-optimal evaluation scores slow down IBO-HPC-** Conditioning on the evaluation score of high-performing configurations is crucial for the performance of IBO-HPC. To analyze the effect of conditioning on evaluation scores of sub-optimal configurations, we conditioned on the $\{0.25, 0.5, 0.75\}$-quantile of all evaluation scores obtained until iteration $t$. As expected, for higher quantiles (i.e. better evaluation scores), IBO-HPC finds better configurations.

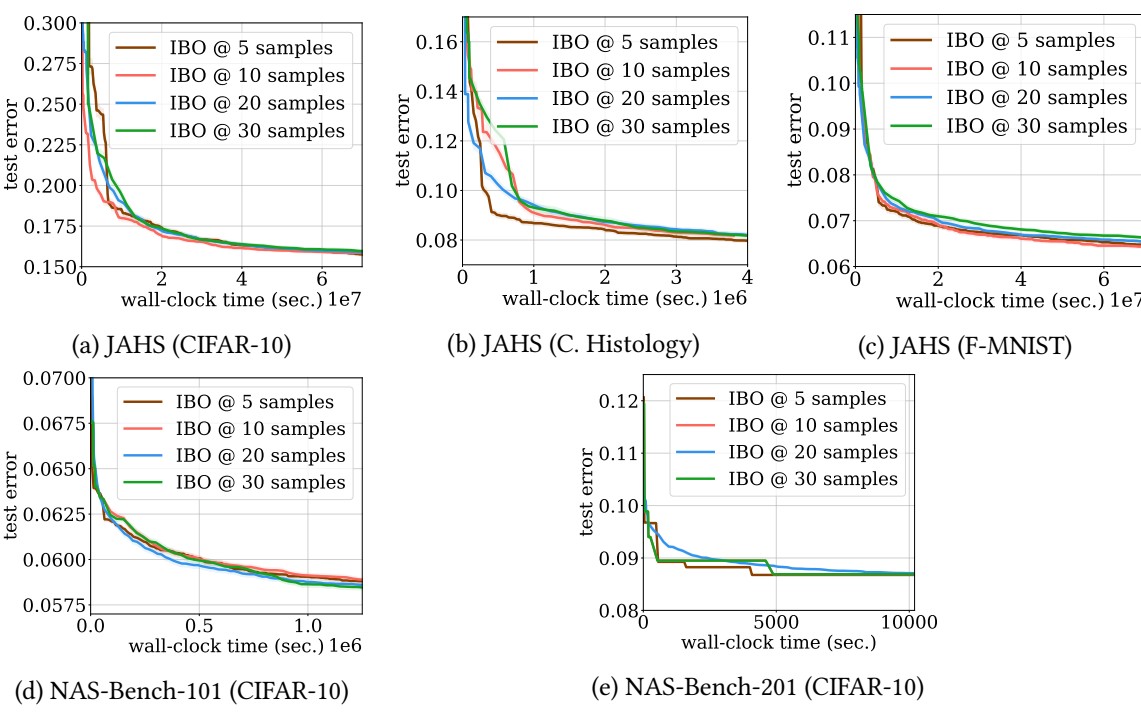

(a) JAHS (CIFAR-10)    (b) JAHS (C. Histology)    (c) JAHS (F-MNIST)

(d) NAS-Bench-101 (CIFAR-10)    (e) NAS-Bench-201 (CIFAR-10)

Figure 13: **$L$ has no significant effect on IBO-HPC's performance**. We found that fixing the surrogate model for $L = \{5, 10, 20, 30\}$ iterations does not lead to significant differences in the performance and convergence speed of IBO-HPC. Only in earlier iterations was a significant variation in convergence speed found on JAHS CIFAR-10 and JAHS CO. However, these variations vanish with the progress of optimization.

## E.5 Statistical Significance

We applied a one-sided Wilcoxon test to validate our results are statistically significant. Tab. 2 provides p-values for comparing IBO-HPC against $\pi$BO, BOPrO, and Priorband for runs in which the user interaction happened at the 5th iteration. Tab. 3 shows the same for runs with user interactions at the 10th iteration. Note that we only provided user interactions at the 10th iteration for JAHS, NAS-201, and NAS-101. Further, we compare IBO-HPC with BO w/ RF, BO w/ TPE, and SMAC. In case of runs without user interaction (see Tab. 4). Overall, it can be seen that IBO-HPC significantly outperforms $\pi$BO, BOPrO, and Priorband if user knowledge is provided. Also, there is no clear pattern in terms of significance when comparing IBO-HPC with other BO methods when no interaction takes place. In approximately 50% of the cases, IBO-HPC outperforms the baselines. In the other cases, the baselines outperform IBO-HPC. Thus, we conclude that IBO-HPC is competitive when no user knowledge is given. We set a significance level of $p = 0.05$.

| | IBO-HPC vs. PiBO | BO-HPC vs. BOPrO | BO-HPC vs. Priorband |
|---|---|---|---|
| JAHS (CIFAR10) | $\mathbf{2.0 \times 10^{-9}}$ | $9.0 \times 10^{-1}$ | $\mathbf{8.8 \times 10^{-16}}$ |
| JAHS (C. Hist.) | $\mathbf{3.0 \times 10^{-7}}$ | $\mathbf{1.5 \times 10^{-3}}$ | $\mathbf{8.9 \times 10^{-16}}$ |
| JAHS (F.-MNIST) | $\mathbf{1.0 \times 10^{-2}}$ | $\mathbf{1.2 \times 10^{-2}}$ | $\mathbf{1.8 \times 10^{-15}}$ |
| NAS201 | $\mathbf{1.9 \times 10^{-6}}$ | $9.6 \times 10^{-1}$ | $1.4 \times 10^{-1}$ |
| NAS101 | $\mathbf{1.3 \times 10^{-4}}$ | $\mathbf{2.8 \times 10^{-4}}$ | $\mathbf{2.4 \times 10^{-4}}$ |
| HPO-B (6767:31) | $98 \times 10^{-1}$ | $6.0 \times 10^{-2}$ | $\mathbf{2.6 \times 10^{-5}}$ |
| HPO-B (6794:31) | $\mathbf{2.0 \times 10^{-2}}$ | $\mathbf{5.2 \times 10^{-6}}$ | $\mathbf{1.5 \times 10^{-13}}$ |
| HPO-B (6794:9914) | $9.9 \times 10^{-1}$ | $9.9 \times 10^{-1}$ | $\mathbf{2.0 \times 10^{-2}}$ |
| PD1 (Imagenet) | $\mathbf{8.9 \times 10^{-16}}$ | $\mathbf{3.6 \times 10^{-15}}$ | - |
| FCNet (Slice Localization) | $\mathbf{3.0 \times 10^{-4}}$ | $\mathbf{1.8 \times 10^{-3}}$ | $\mathbf{1.9 \times 10^{-7}}$ |

Table 2: **IBO-HPC significantly outperforms $\pi$BO, BOPrO and Priorband**. IBO-HPC significantly outperforms our baselines that allow for user priors. The table above shows p-values of the Wilcoxon test with significance level $p = 0.05$ for runs in which the same beneficial user knowledge was provided to all algorithms. For IBO-HPC, the knowledge was provided at the 5th iteration, while for the baselines, the knowledge was provided ex ante. Significance is reported in **bold**.

| | IBO-HPC vs. PiBO | IBO-HPC vs. BOPrO | IBO-HPC vs. Priorband |
|---|---|---|---|
| JAHS (CIFAR10) | $\mathbf{1.6 \times 10^{-10}}$ | $9.9 \times 10^{-1}$ | $\mathbf{1.7 \times 10^{-15}}$ |
| JAHS (C. Hist.) | $\mathbf{1.2 \times 10^{-7}}$ | $\mathbf{1.1 \times 10^{-3}}$ | $\mathbf{1.7 \times 10^{-15}}$ |
| JAHS (F.-MNIST) | $\mathbf{8.9 \times 10^{-3}}$ | $8.2 \times 10^{-2}$ | $\mathbf{1.8 \times 10^{-15}}$ |
| NAS201 | $\mathbf{1.9 \times 10^{-6}}$ | $9.9 \times 10^{-1}$ | $1.0 \times 10^{-1}$ |
| NAS101 | $\mathbf{1.3 \times 10^{-4}}$ | $\mathbf{1.3 \times 10^{-4}}$ | $\mathbf{2.9 \times 10^{-4}}$ |

Table 3: **IBO-HPC significantly outperforms $\pi$BO, BOPrO and Priorband**. IBO-HPC significantly outperforms our baselines that allow for user priors. The table above shows p-values of the Wilcoxon test with significance level $p = 0.05$ for runs in which the same beneficial user knowledge was provided to all algorithms. For IBO-HPC, the knowledge was provided at the 10th iteration, while for the baselines, the knowledge was provided ex ante. Significance is reported in **bold**.

|  | IBO-HPC vs. BO /w RF | IBO-HPC vs. BO /w TPE | IBO-HPC vs. SMAC |
|---|---|---|---|
| JAHS (CIFAR10) | $5.3 \times 10^{-9}$ | 0.19 | 0.9 |
| JAHS (C. Histology) | $9.8 \times 10^{-6}$ | 0.93 | 0.96 |
| JAHS (Fashion-MNIST) | 0.08 | 0.28 | 0.52 |
| NAS201 | $1.4 \times 10^{-4}$ | $7.8 \times 10^{-4}$ | $7.7 \times 10^{-3}$ |
| NAS101 | $8.8 \times 10^{-4}$ | $9.5 \times 10^{-7}$ | $2.8 \times 10^{-4}$ |
| HPO-B (6767:31) | $3.1 \times 10^{-4}$ | 0.31 | $6.7 \times 10^{-4}$ |
| HPO-B (6794:31) | 0.98 | **0.01** | 0.99 |
| HPO-B (6794:9914) | 0.99 | $1.3 \times 10^{-5}$ | 0.38 |
| PD1 | $8.9 \times 10^{-16}$ | $8.9 \times 10^{-16}$ | $8.8 \times 10^{-16}$ |
| FCNet | 0.97 | **0.01** | 0.99 |

Table 4: **IBO-HPC is competitive with BO baselines**. IBO-HPC significantly outperforms our baselines in 50% of the cases when no user knowledge is provided. The table above shows p-values of the Wilcoxon test with significance level $p = 0.05$ for runs in which no user knowledge was provided. It can be seen that no clear pattern is recognizable. Hence, there is no clear winner among the competing algorithms on standard HPO tasks. Thus, IBO-HPC can be seen as competitive in these settings. Significance is reported in **bold**.

### E.6 Cost Efficiency of IBO-HPC's Selection Policy

We now provide details on the computational costs of IBO-HPC. Therefore, we analyzed the composition of the overall runtime of an optimization run and measured the time needed to train configurations suggested by IBO-HPC versus the time spent on actually performing optimization (including fitting the surrogate PC and sampling new configurations). Fig. 14a shows that the computation time spent on learning the PC and sampling new configurations is negligible compared to the time spent on training the suggested configurations. Additionally, 14b shows that IBO-HPC is faster than SMAC in 4/5 cases in terms of runtime. Here, we considered the time spent in updating the surrogate and suggesting new configurations. Note that this does not include training costs. Interestingly, with the increasing size of the search space, the efficiency advantage of IBO-HPC is increasing. We suspect that the intensify-mechanism in SMAC, which includes a local search, is the reason for the higher computational costs of SMAC.

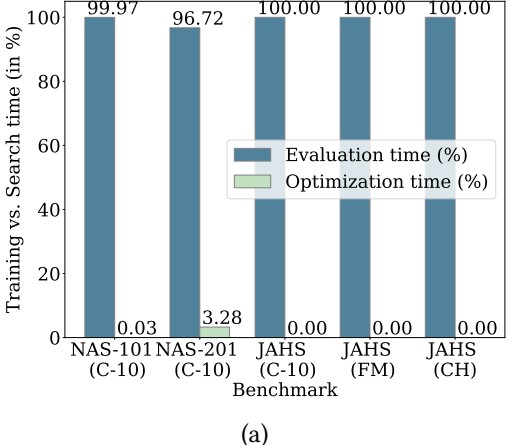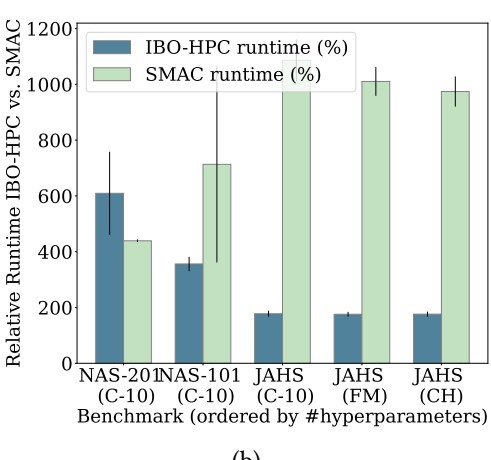

(a)                                                                  (b)

Figure 14: **IBO-HPC is a cost-efficient HPO method**. (a) Learning a surrogate and suggesting new configurations is negligible in terms of computational costs compared to training the suggested configurations. We computed the time spent on training configurations (blue) vs. time spent learning a PC and suggesting new configurations (orange). In all experiments, the training of configurations caused the large majority of computational costs, often even approaching 100%. (b) IBO-HPC is more efficient than the prominent HPO algorithm SMAC in 4/5 cases (averaged over 20 runs). Also, with the increasing number of hyperparameters, the gap between IBO-HPC and SMAC in terms of computational efficiency is larger. We report runtimes normalized between [0, 1] per benchmark s.t. the highest obtained runtime for a given benchmark is 1.

### E.7 Exploration-Exploitation Trade-off of IBO-HPC

An effective mechanism to trade off exploration versus exploitation is crucial for high-performing hyperparameter optimization algorithms. Below we show that IBO-HPC's sampling policy effectively achieves this trade-off. In early iterations, IBO-HPC explores the search space (high sample variance), while in later iterations, it exploits the knowledge collected (low sample variance).

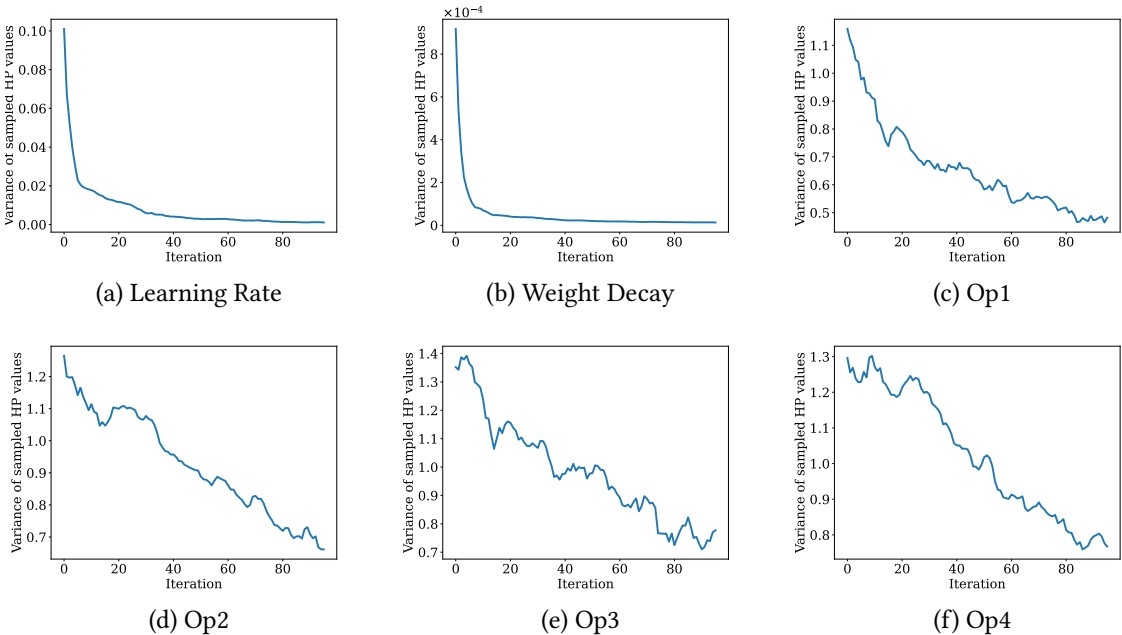

Figure 15: **IBO-HPC effectively trades off exploration and exploitation**. IBO-HPC's sampling policy naturally and effectively trades off exploration (high sampling variance in early iterations) versus exploitation (low sampling variance in later iterations). We show the sampling variance of 6 hyperparameters of the JAHS benchmark (CIFAR10) for each iteration, averaged over 20 runs of IBO-HPC.

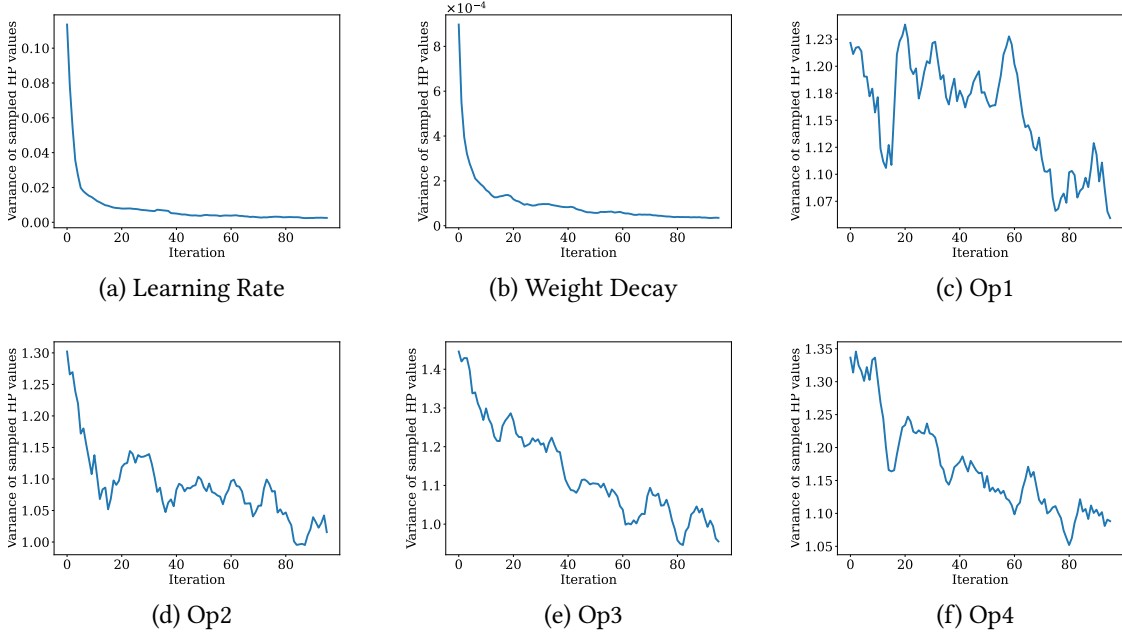

Figure 16: **IBO-HPC effectively trades off exploration and exploitation**. IBO-HPC's sampling policy naturally and effectively trades off exploration (high sampling variance in early iterations) versus exploitation (low sampling variance in later iterations). We show the sampling variance of 6 hyperparameters of the JAHS benchmark (Colorectal Histology) for each iteration, averaged over 20 runs of IBO-HPC.

### E.8 Hyperparameters of IBO-HPC

IBO-HPC comes with a few hyperparameters itself, which have to be set. For our experiments, we set the number of iterations the surrogate is kept fixed $L = 20$, the decay value $\gamma = 0.9$. We let all methods optimize for 2000 iterations for fair comparison. Our surrogate models, i.e., PCs and the associated learning algorithm, have some hyperparameters as well. The structure learning algorithm splits use the RDC independence test and K-means clustering. The threshold to detect independencies is set to 0.3, and the minimum number of instances per leaf is adapted dynamically based on the number of configurations tested during an optimization run.

### E.9 Hardware and Computational Cost

We ran all our experiments on DGX-A100 machines and used 10 CPUs for each run, thus parallelizing some sub-routines (e.g. learning of PCs). We did not use any GPUs as we queried the benchmarks employed to provide the performance of configurations. The JAHS benchmark requires a relatively large RAM ($> 16GB$) to run smoothly as it loads large ensemble models.

**Computational Cost.** We used HPO and NAS benchmarks to provide reproducible results and to keep the computational effort as low as possible, allowing researchers with relatively low computational resources to reproduce our results. Considering all baselines and all IBO-HPC runs, we recorded approximately 70k algorithm executions. On average, one run lasts 15 minutes (thanks to the benchmarks), resulting in approximately 1800 CPU hours (on DGX-A100 machines) needed for our experimental evaluation. Note that due to the use of benchmarks, we did not need any GPUs.

Note that these computational costs reflect the cost of running all the HPO algorithms and are not to be confused with the computational costs reported by the benchmarks. In contrast, the benchmarks provide a wall-clock time estimate of training and evaluating a configuration from a hyperparameter search space. This allows us to plot the test error against the computation time.

## F  Limitations and Future Work

IBO-HPC allows users to act as an external source of knowledge that can help to solve HPO tasks more efficiently. While this is an important step towards a more inclusive vision of AutoML, IBO-HPC also ignores another crucial source of information, namely previous HPO runs. Since IBO-HPC is built on PCs and PCs are a modular architecture, one could leverage PCs – learned on previous HPO runs – to act as a guide for future HPO tasks. This way, one could incorporate user knowledge *and* information from previous HPO runs to increase the efficiency of HPO.

