# OpenReview forum: "Hyperparameter Optimization via Interacting with Probabilistic Circuits"
_automl.cc/AutoML/2025/Methods_Track — AutoML 2025 Methods Track_

### Official Review · Reviewer_5mV6 · 2025-04-29

**Comments To Authors:**

### A summary of the contributions

This paper introduces IBO-HPC (Interactive Bayesian Optimization via Hyperparameter Probabilistic Circuits), a new approach for hyperparameter optimization that directly incorporates user knowledge during the optimization process. The main contributions include: (1) a novel BO method using probabilistic circuits as surrogate models that eliminates inner-loop acquisition function optimization, (2) theoretical convergence analysis of the method, and (3) empirical evidence demonstrating effectiveness against state-of-the-art HPO methods. The authors compare their work with prior approaches like BOPrO and cBO, highlighting their key innovation of directly conditioning the surrogate on user beliefs rather than weighting acquisition functions. The authors show that BO-HPC is Competitive in HPO & NAS and that the method takes advantage of helpful user inputs but is resiliant to misleading suggestion.

### Potential impact on the field of AutoML

This paper addresses a significant gap in AutoML by enabling an interactive hyperparameter optimization. It will likely have a positive impact for several reasons:
- It advances the notion of "human-centered AutoML" by enabling experts to effectively guide optimization during the process. This solves a practical problem of making sure HPO improves on human knowledge instead of replacing it.
- The novel use of probabilistic circuits provides a principled way to incorporate user feedback that accurately reflects user beliefs
- The approach could spark follow-up research on extending interactive capabilities to other AutoML tasks, developing more sophisticated user interfaces for knowledge elicitation, and exploring how to leverage PCs for transfer learning across related tasks

### Technical quality and correctness

The paper demonstrates high technical quality through both theoretical analysis and empirical validation. The claims are supported by:
- Formal proofs showing that IBO-HPC is a feedback-adhering interactive policy; Proposition 1 establishes that the policy is both efficacious (changed by user knowledge) and feedback-adhering (accurately reflects user beliefs)
- Proof that IBO-HPC is a global optimizer; Proposition 2 provides a lower bound on the expected improvement in each iteration.
- Analysis of convergence behavior (Proposition 3)
- Comprehensive experiments across 10 tasks from 6 benchmarks with proper ablation studies examining the impact of diferent components.

### Clarity of the contributions

The paper is well-structured and clearly presented. The methodology is explained systematically with clear algorithm descriptions and formal definitions. However, more intuitive explanation of Probabilistic Circuits can be helpful to ensure adoption among practicioners. The experimental section covers key research questions with comprehensive plots and analyses. The figures, especially Figure 1, effectively illustrate the key concepts and advantages of the proposed approach.

### Potential ethical concerns

No significant ethical concerns are apparent with this work.

### Final assessment

Strengths:
- Novel integration of probabilistic circuits in HPO providing tractable conditioning on user knowledge
- Theoretical guarantees on the method's properties (feedback-adherence, global optimization, convergence)
- Demonstrated ability to recover from misleading user knowledge
- Significant speedups (2-10×) when beneficial user knowledge is provided
- Competitive performance even without user interactions

Weaknesses:
- The method introduces its own hyperparameters (like decay rate ω) that require tuning
- Probabilistic circuits add complexity compared to simpler surrogate models
- Limited discussion on how users might formulate knowledge in practice
- No evaluation with actual human users (though this is common)

The contributions are highly valuable as they address a meaningful gap in HPO - incorporating expert knowledge interactively during optimization, not just beforehand. The method is theoretically sound and empirically effective, offering a new direction for human-in-the-loop AutoML.

### Final Recommendation

The paper presents significant innovations in interactive HPO with solid theoretical foundations and compelling empirical results. It addresses an important practical problem and opens promising research directions in human-centered AutoML.

**Review Confidence:**

3

**Review Rating:**

8

---

### Official Review · Reviewer_wsWS · 2025-04-30

**Comments To Authors:**

**Summary**: This paper introduces Interactive Bayesian Optimization via Hyperparameter Probabilistic Circuits (IBO-HPC), a framework that builds probabilistic circuits (PCs) as the surrogate model for BO. PCs allow users to directly condition the surrogate on their beliefs to interact with the BO optimization process. Results show that IBO-HPC achieves state-of-the-art performance on both standard HPO and interactive HPO problems.

**Strength**: Applying PCs to BO is novel and interesting. The evaluation results show the state-of-the-art performance on different benchmarks. Additionally, IBO-HPC shows that it could handle different types of feedback from humans.

**Weaknesses**: During the optimization process, this method introduces many hyperparameters that are not applied in other BO methods (And anytime, there is no good intuition for that). For instance, the authors only fit PCs in every L-th iteration. Meanwhile, the model is updated in other BO frameworks after each iteration. The authors provide an ablation study in Figure 13. However, I would also like to see the results for L=1. Additionally, it is unclear how the number of conditions N (line 8, algorithm 1) is set and what the impact of this value is. These values can also be

* Although the authors state that their sampling policy differs from TS. However, since both approaches are sampling-based, the author should also compare IBO-HPC with TS. For instance, since equation 2 does not specify the distribution form, one can also apply a GP to do the sampling.

**Minor Points**
* The figure in the paper is not clearly illustrated. For instance, the meaning of (BO (w/ dist. intervention@5) is a bit unclear for me in Figure 2 (the cyan and orange lines are also a bit confusing. However, at least they are explained in the caption.
* It would also be interesting to see the sampling trajectory of IBO-HPC after the human interaction
* No uncertainty is given in plots despite the fact that multiple runs are evaluated.


**Questions** I have several questions while reading the paper:
* How do different HPO methods set their initial points? From Figure 3, the test error from different optimizers already differs even before the first misleading interaction. It looks like the initialization strategy has a great impact on the optimization process.
* Why does IBO (w/ interaction@10) already achieve a test error in Figure 2 (a) and (b) even before the second interaction is inserted (especially in Figure b)?

**Review Confidence:**

3

**Review Rating:**

6

---

### Official Review · Reviewer_PJBS · 2025-05-01

**Comments To Authors:**

# Summary

The authors propose IBO-HPC, Bayesian Optimization (BO) with interactive probabilistic circuits (PCs). With PCs, users can specify a prior over hyperparameters (HPs) (can be any prior, normal distribution or setting a HP to a certain value) at any time in the optimization process. PCs encode distributions over random variables as a graph. Because of this, no acquisition function is needed to propose the next candidate to evaluate, as candidates are directly sampled from the PC. Over time, the PC becomes more exploitative with more data. The authors evaluate their method on selected tasks from HPO & NAS benchmarks and show that BO with PCs performs on par with SOTA BO methods, and IBO-HPC with good priors performs better than previous baselines considering user priors. IBO-HPC with bad priors has the ability to recover.

# Relation to SOTA

The authors seem to have considered previous work in a comprehensive way.

# Correctness of Results

The results seem to be correct.

# Strengths and Weaknesses of the Paper

## Strengths

Elegant idea of incorporating user priors at any time via an alternative surrogate model, namely Probabilistic Circuits. Thorough evaluation of every aspect of the method, i.e. its hyperparameters, runtime, comparison of results per task and optimizer. Very clear presentation.

## Weaknesses

How the tasks were selected is a bit unclear, it is unclear what ‘diverse’ means in this context.
Minor comments listed below.

# Relevance

The contributions of this paper are relevant because it provides a new paradigm to introduce user priors in BO. Using a PC as a surrogate is also an interesting avenue.

# Final Recommendation

Accept.

# Additional Feedback

* Baselines: L246: If you have used the SMAC3 package, also cite \[Lindauer et al., 2022\]. In that case I also guess that the random forest surrogate model has been used.
* Clarity/Accessibility: (e.g. Fig. 2\) The colors used throughout the figures are inconsistent as well as the method naming. In addition, you should consider using a colorblind palette and saturated colors. I.e., BOPrO and RS are barely distinguishable. You can also consider using markers or increasing the linewidth.
* Task selection: L254: It is unclear how you have selected tasks. What does diverse mean? For example, there are more tasks in PD1. It could also be nice to provide a tabular task overview in the appendix, i.e. with search space metadata.
* L259: Did you really use 500 seeds? Why, were the variances so big? In that case, I would have some general concerns. Or is this a typo?
* User Priors: L263: With that sampling approach for configurations, how do you select which HP gets a prior? Or does every HP get one? This part is unclear to me and also did not become clearer after Appendix E.3.
* Limitations of the method: The limitations seem to be a bit thin (no method has it all), if there are some, I would like to see them more explicitly. If there are none, even better for the method. E.g. these results hold for HPO & NAS tasks and we do not know how it translates to other domains.
* Just comments from my side:
  * It is interesting that local search sometimes performs very well.
  * It is also interesting that the exploration-exploitation trade-off is quasi preset to go from exploration to exploitation.
  * It would also be interesting to study PCs as a surrogate model in general.

\[Lindauer et al., 2022\] Lindauer, M., Eggensperger, K., Feurer, M., Biedenkapp, A., Deng, D., Benjamins, C., ... & Hutter, F. (2022). SMAC3: A versatile Bayesian optimization package for hyperparameter optimization. *Journal of Machine Learning Research*, *23*(54), 1-9.

**Review Confidence:**

4

**Review Rating:**

8

---

### Meta-Review · Area_Chair_42Wt · 2025-05-06

**Recommendation:** Accept
**Confidence:** 4

**Metareview:**

paper proposes to include probabilistic circuits as a surrogate model for BO which allows to perform exact conditional inference and sampling. This allows to include user priors and eliminate the need of inner optimization in BO. Reviewers noted that the paper is well written and experiments shows good improvement over the considered baselines. Finally, the reproducibility reviewer also commended the paper. For those reasons, I recommend accepting the paper.